# Spike-in enhanced phosphoproteomics uncovers synergistic signaling responses to MEK inhibition in colon cancer cells

Mirjam van Bentum[1,2], Bertram Klinger [2,3], Anja Sieber [2,3], Sheyda Naghiloo [1], Henrik Zauber[1], Nadine Lehmann [3], Mohamed Haji[1], Sylvia Niquet[1,4], Philipp Mertins [1,4], Nils Blüthgen [2,3] ✉ & Matthias Selbach [1,3] ✉

Targeted kinase inhibitors are a cornerstone of cancer therapy, but their success is often hindered by the complexity of cellular signaling networks that can lead to resistance. Overcoming this challenge necessitates a deep understanding of cellular signaling responses. While standard global phosphoproteomics offers extensive insights, lengthy processing times, the complexity of data interpretation, and frequent omission of crucial phosphorylation sites limit its utility. Here, we combine data-independent acquisition (DIA) with spike-in of synthetic heavy stable isotope-labeled phosphopeptides to facilitate the targeted detection of particularly informative phosphorylation sites. Our spike-in enhanced detection in DIA (SPIED-DIA) approach integrates the improved sensitivity of spike-in-based targeted detection with the discovery potential of global phosphoproteomics into a simple workflow. We employed this method to investigate synergistic signaling responses in colorectal cancer cell lines following MEK inhibition. Our findings highlight that combining MEK inhibition with growth factor stimulation synergistically activates JNK signaling in HCT116 cells. This synergy emphasizes the therapeutic potential of concurrently targeting MEK and JNK pathways, as evidenced by the significantly impaired growth of HCT116 cells when treated with both inhibitors. Our results demonstrate that SPIED-DIA effectively identifies synergistic signaling responses in colorectal cancer cells, presenting a valuable tool for uncovering new therapeutic targets and strategies in cancer treatment.

Cell signaling plays a key role in human health and disease, and sustained proliferative signaling is recognized as a hallmark of cancer[1]. Correspondingly, kinase inhibitors have established their crucial role in the arsenal of cancer therapy, demonstrating significant efficacy in targeting these proliferative pathways[2,3]. However, cellular responses to targeted therapies are often complicated by resistance mechanisms.

Primary resistance to targeted treatment can be caused by feedback mechanisms that lead to rewiring or reactivation of signaling pathways. For example, resistance to PI3K/mTOR inhibition in breast cancer is frequently due to feedback mechanisms that cause activation of JAK/STAT signaling[4]. In neuroblastoma, resistance to Mitogen-Activated Protein Kinase Kinase (MEK) inhibitors can emerge from

[1]Max Delbrück Center for Molecular Medicine, Robert-Rössle-Str. 10, 13092 Berlin, Germany. [2]Institute for Theoretical Biology, Faculty of Life Sciences, Humboldt-Universität zu Berlin, Unter den Linden 6, 10099 Berlin, Germany. [3]Charité-Universitätsmedizin Berlin, Charitéplatz 1, 10117 Berlin, Germany. [4]Berlin Institute of Health, Berlin, Germany. ✉e-mail: nils.bluethgen@charite.de; matthias.selbach@mdc-berlin.de

negative feedback mechanisms within the MAPK signaling and via the IGF receptor, thereby reactivating MAPK signaling upon treatment[5]. In colorectal cancer (CRC), therapy targeting the MAPK pathway is undermined by negative feedback, which results in increased sensitivity of the EGF receptor and consequently leads to the reactivation of both MAPK and AKT signaling pathways[6,7]. These studies collectively emphasize the importance of characterizing cell signaling to predict the outcome of targeted treatment.

Genomic and transcriptomic markers often cannot predict cell line specific resistance to targeted therapy, highlighting the need for alternative methods to characterize cell signaling[5,8,9]. Mass spectrometry-based phosphoproteomics is arguably the best available technology to comprehensively characterize cellular signaling states[10–12]. However, some kinases and phosphorylation sites particularly informative about critical cellular signaling states are low abundant, requiring considerable phosphoproteomics depth to enable their detection. In addition, correct biological interpretation of phosphoproteomic data hinges on the precise and accurate quantification of identified phosphopeptides.

Classical global phosphoproteomics using data-dependent acquisition (DDA) can routinely identify thousands of phosphorylation sites in a single sample, with more than 200,000 human phosphorylation sites being mapped in total[13–15]. To attain adequate coverage, DDA global phosphoproteomics relies on extensive fractionation, resulting in long data acquisition times. Despite these efforts, specific phosphorylation sites of interest can still remain undetected. Data-independent acquisition (DIA) has emerged as an attractive alternative to classical DDA due to greater proteome coverage of single shot analyses, significantly increasing throughput[16,17]. In combination with ion mobility-based peptide fractionation in the gas phase (diaPASEF), this also results in higher sensitivity[18]. DIA has also been applied to phosphoproteomics[19–22]. However, the number of unique phosphorylation sites identified by single shot DIA phosphoproteomics is typically lower than in classical global DDA phosphoproteomics, aggravating the problem of missing specific phosphosites of interest[23]. Additionally, due to the highly complex spectra of DIA runs, false discovery control remains an ongoing point of discussion.

An attractive alternative to global phosphoproteomics is the targeted detection of a limited set of key phosphorylation sites reflecting critical kinases and substrates pivotal to cell signaling. Indeed, targeted (phospho-)proteomics approaches such as selected reaction monitoring (SRM) and parallel reaction monitoring (PRM) enable fast, sensitive, and reproducible detection of target peptides and have been successfully applied to study signaling[24–27]. More recently, a number of advanced targeted acquisition methods have been described that employ more sophisticated acquisition strategies to increase the efficiency of mass spectrometers in detecting target peptides[28]. For example, spike-in triggered acquisition methods enabled targeted detection of several hundred tyrosine phosphorylated peptides commonly dysregulated in cancer[29]. While these methods significantly enhance the sensitivity and consistency of target phosphopeptide detection, they demand extensive method development for each target peptide and often involve partly manual data analysis, which can be time-consuming. This complexity limits their practicality for high-throughput experiments. Additionally, while targeted approaches offer precise detection of specific phosphopeptides, they inherently limit the broader discovery potential that is a hallmark of global DDA and DIA phosphoproteomics. Recently, DIA was combined with spike-in triggered acquisition, enabling integrated targeted and DIA-based discovery phosphoproteomics[30]. However, this workflow also requires extensive method optimization and is only available on specific mass spectrometers. Adding an excess of a heavy stable isotope-labeled reference has been shown to increase sensitivity in DIA[31–33]. The use of heavy stable-isotope labeled synthetic spike-in peptides is also well-established in targeted proteomics and has been shown to increase confidence in target identification and the accuracy of quantification[34–36].

Here, we combine these two concepts to develop *spike-in enhanced detection in DIA* (SPIED-DIA) as a simple and generic method to improve detection of key phosphopeptides in DIA phosphoproteomics. To this end, we synthesize a custom set of heavy stable isotope-labeled phosphopeptides covering a wide range of signaling pathways. We show that spiking-in this heavy stable isotope labeled reference set improves detection and quantification of key target phosphorylation sites up to three fold. At the same time, the method takes full advantage of the discovery potential of conventional DIA. Applying SPIED-DIA to CRC cells reveals that MEK inhibition stimulates growth factor-induced JNK signaling in HCT116 cells. Consistently, we observe that combinatorial treatment of this cell line with MEK and JNK inhibitors synergistically impairs growth. Hence, phosphorylation-based signaling responses identified by SPIED-DIA can identify effective drug combinations that overcome primary resistance.

## Results
### Spike-in enhanced detection improves sensitivity of DIA phosphoproteomics

Heavy stable isotope-labeled synthetic peptides are widely used for targeted (phospho-)proteomics. However, currently available methods either require extensive optimization of acquisition methods, SRM or PRM[28], or complex DDA methods, spike-in triggered acquisition methods such as TOMAHAQ or Surequant[29,37]. We hypothesized that combining a heavy stable isotope-labeled spike-in with standard DIA could leverage the simplicity of DIA along with the enhanced sensitivity provided by the heavy spike-in (Fig. 1A). Specifically, spiking-in a set of heavy synthetic phosphopeptides into a complex proteome sample would both generate (i) global untargeted phosphoproteomic data and (ii) improve coverage of the targeted phosphopeptides. The increased sensitivity for targeted phosphopeptides is based on the idea that the heavy spike-in peptides serve as beacons facilitating the detection of corresponding endogenous, light, phosphopeptides: While low abundant endogenous phosphopeptides might escape detection, their more abundant synthetic heavy peptide counterpart spiked-in in excess are more readily detectable. This enables identification of the correct retention time and ion mobility of the corresponding light endogenous counterpart, facilitating its detection (Fig. 1B). Additionally, the same heavy peptides spiked into different samples serves as an internal reference improving across-sample quantification (by computing the ratio of the within sample light to heavy ratios). Hence, spike-in peptides serve a dual purpose: improvement of detection rate and quantitative performance. Importantly, when applying the final workflow, a single DIA raw data file is used to perform two distinct types of analyses: one that generates global untargeted label-free quantified data and another, the SPIED-DIA analysis, which allows for more sensitive detection and stable isotope-based quantification of targeted peptides.

We tested this idea by measuring the global phosphoproteome of mixed heavy (H) and light (L) HCT116 cell lysates generated using stable isotope labeling with amino acids in cell culture (SILAC)[38] in serial dilution (Fig. 1C). Following phosphopeptide enrichment, samples were measured on a timsTOF Pro2 mass spectrometer in triplicates before being processed with DIA-NN[16,17,31] to identify 12,621 unique phosphorylation sites. These data were first analyzed using the standard workflow (normal SILAC-DIA). Detected median peptide ratios were close to expected values, indicating accurate quantification (Fig. 1D). As expected, we observed a rapid reduction in the number of quantifiable precursor ratios throughout the dilution series, with essentially no target precursors detected at 1:63 dilution or higher. Next we took advantage of the reference peptides as beacons to

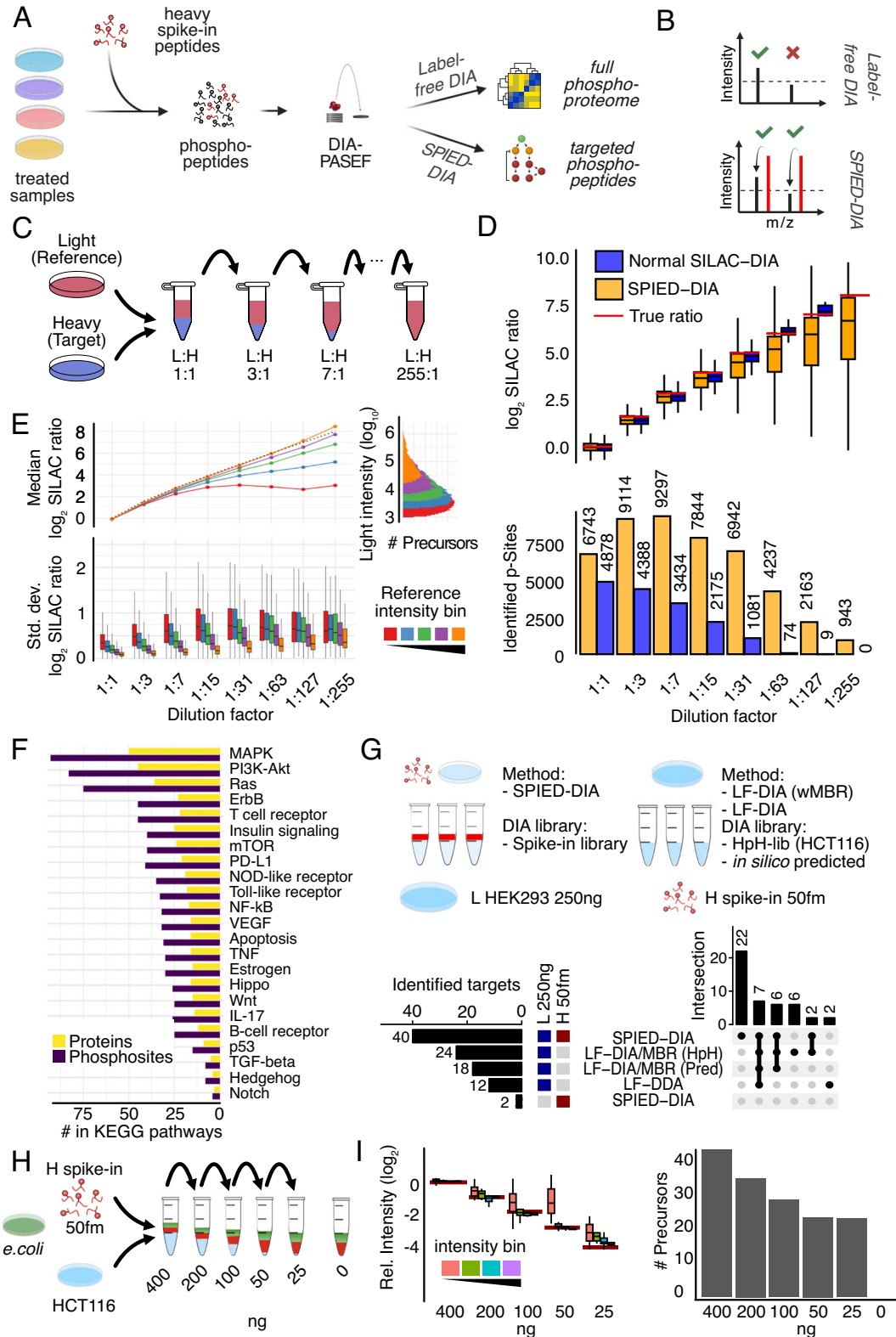

facilitate detection of target peptides, whereby only confident identification of phosphopeptides in the reference channel was required, relaxing precursor confidence threshold for the target channel (see Material and Methods for details). We note that in this experiment, we are using all the SILAC lights as references for all the corresponding SILAC heavies. For each precursor that passed the combined confidence threshold, we extracted intensities in the target channel. Similar to requantify in MaxQuant[39], RefQuant for dimethyl labeling[32] and the requantify option in DIA spike-in SILAC[33], this spike-enhanced detection approach conceptually uncouples detection and quantification. In this way, it provides intensity information for target channel precursors that would otherwise escape detection.

Fig. 1 | Design and benchmarking of Spike-In Enhanced Detection in Data-Independent Acquisition (SPIED-DIA). Spike-in enhanced detection in data independent acquisition (SPIED-DIA) (on previous page) (**A**) Workflow depicting the integration of Label-free (LF) DIA with SPIED-DIA. **B** Quantification and identification in LF-DIA and SPIED-DIA. **C** Schematic of the dilution series used to benchmark the performance of SPIED-DIA. **D** Number of phospho-sequences identified in at least 2 out of 3 technical replicates and corresponding H/L channel intensities. **E** SILAC ratio variability by light intensity and dilution derived from all quantified precursors in three technical replicates. Lines show median log2 SILAC ratios across dilution factors, binned by reference channel (L) intensity with equally sized bins. Expected ratios are shown as a dotted line. Histogram depicts precursor count distribution by log10 light intensity. Boxplots show standard deviation of log2 SILAC ratios by dilution and bin, aggregated at the modified sequence level. **F** Signaling pathway origins of phosphosites selected for spike-in peptide library. **G** Improved target peptide identification with SPIED-DIA. Targets identified in 2 out of 3 biological replicates, with a CV lower than 10%. Upset plot depicts intersections > 1. wMBR: with match-between-runs, HpH-lib: library created by high pH fractionation phosphoproteomics. **H** Dilution series of heavy peptides in light samples in 100 ng *E. coli* background, 3 replicates. **I** Relative mean intensity ($n = 3$) of precursors normalised to 400 ng L condition, filtered as in panel **G** Red lines depict expected relative intensity. All Boxplots show the median, interquartile range, whiskers at max 1.5×IQR, and exclude outliers. Source data are provided as a Source Data file.

We found that spike-in enhanced detection greatly increased the number of quantifiable phosphopeptides in the target channel even at high dilutions (Fig. 1D). As expected, the data obtained in this manner showed a greater spread (lower precision) and signs of systematic ratio compression at higher dilutions (lower accuracy). To investigate this further, we binned precursors by their intensities and plotted observed SILAC ratios as a function of the dilution series (Fig. 1E). While precursors with high intensities accurately reflected the expected ratios up to a dilution of 1:255, low intensity precursors started to saturate at dilutions of 1:15. Importantly, despite this ratio compression, the observed ratios consistently showed the correct trend. We also compared SILAC ratios for precursors quantified in both the standard and the SPIED-DIA workflow and observed high consistency (Supplementary Fig. 1). In conclusion, precursors identified by both the standard and the SPIED-DIA workflow generated similar ratios. However, SPIED-DIA markedly improved coverage, revealing many additional ratios with varying degrees of ratio compression.

Since peptides with more than one serine, threonine and tyrosine residue can be phosphorylated at different positions, we also tested the quality of phosphorylation site localisation. To this end, we compared H/L ratios for identical heavy and light pairs (that is, peptides phosphorylated on the same amino acid), and positional isomers (that is, peptides with identical amino acid sequence but phosphorylated at different sites). We observed higher precision for pairs than for isomers, especially when filtering for post-translational modification (PTM) site confidence, indicating that DIA-NN accurately localises phosphorylation sites (Supplementary Fig. 1). We further assessed phospho-site localisation by validating the localisation in spectra from raw files (Supplementary Fig. 2).

### Synthetic phosphopeptide panel to study cell signaling

Encouraged by these results, we sought to apply spike-in enhanced detection to study cell signaling. Specifically, we reasoned that spiking-in a selected set of synthetic heavy stable isotope-labeled phosphopeptides would facilitate detection of their endogenous, light, counterparts. To this end, we selected a panel of phosphosites informative for the activity of a wide range of biologically relevant signaling pathways either manually or based on the PhosphositePlus database (Fig. 1F). From the PhosphoSitePlus database, we selected phosphosites that either had known annotated kinases, influenced enzymatic activity or were present on proteins part of selected KEGG signaling pathways. Selected sites were mapped to an in silico tryptic digest of the human proteome and 485 peptides meeting criteria for synthesis were ordered as SpikeMix™ Peptide Pools. Analysing this heavy synthetic peptide pool via single shot DDA analysis (see Materials and Methods for details) we identified 240 peptides that we used to create a heavy phosphopeptide library (Supplementary Fig. 1, Supplementary Data 1).

Next, to benchmark target peptide detection, we added the heavy synthetic phosphopeptide pool to HEK293 cell lysate, performed phosphopeptide enrichment and measured samples using DIA-PASEF. To investigate the gain in coverage of targeted sites, we compared SPIED-DIA to conventional label-free quantification without the spike-in (Fig. 1G, Supplementary Fig. 1). For the label-free analysis we tested both an in silico predicted spectral library and an empirically determined library obtained from DDA analysis of deeply high-pH HPLC fractionated phosphopeptide samples (see Materials and Methods for details). Comparing both workflows, we observed a marked increase in identified target peptides from 12 to 24 identified targets to 40 identified target phosphosites, while maintaining a false positive rate of 5%, i.e., 2 targets (Fig. 1G). This increase was not only the result of increased sensitivity from the spike-in but also from the ability to use the smaller empirical phosphopeptide library for the search. Using this library for the sample without the spike-in did not yield any identifications (data not shown). Hence, the increased coverage depends on both the library and the increased sensitivity of spike-in enhanced detection.

Next, we looked at the quantitative performance of SPIED-DIA for the panel of selected phosphosites. To this end, we spiked the same amount of the heavy synthetic peptides into decreasing amounts of light human phosphopeptides. As negative control we also added the same amount of *E. coli* to each sample (Fig. 1H). We observed good accuracy and precision throughout the dilution series, as well as a slight loss of target phosphopeptides at higher dilutions (Fig. 1I), consistent with the global data (Fig. 1D, E). In the negative control, no peptides passed our filtering criteria, highlighting the specificity of our workflow (Fig. 1I). Consequently, our approach reliably distinguishes between low-abundance target peptides that are present and those that are truly absent.

Missing values pose a significant challenge in quantifying changes in peptide abundance across conditions, as they can either represent truly absent peptides or peptides that were missed for technical reasons. In both scenarios, the result is an NA value that cannot be used for reliable comparisons across conditions. The inclusion of a heavy spike-in uncouples detection from quantification, providing a key advantage: when the heavy spike-in is consistently detected, it is unlikely that the corresponding light target peptide is missed for purely technical reasons. Instead, the absence of the light peptide can be more reliably attributed to its genuine absence or low abundance. Making more confident absence calls is a key advantage of targeted proteomic methods like SPIED-DIA. We leveraged this feature to improve quantification of target peptides across conditions by rescuing data in cases where the light peptide was not detected but the heavy peptide was. To implement this advantage, we implemented specific filtering criteria: For a peptide to be used for quantification, it needs to be consistently detected as a heavy spike-in across all samples. Additionally, the light target peptide is required to pass filtering criteria in at least two out of three replicates in at least one experimental condition. This ensures that comparisons across samples reflect changes between signal and signal (light target peptide detected in both conditions) or signal and background noise (light target peptide detected in only one condition), but not background noise and background noise (light target peptide not detected in either condition).

To assess the quantitative performance of our pipeline in more detail, we present L/H ratios for all target peptides that were identified in the 400 ng sample across the dilution series (Supplementary Fig. 3). For most peptides, we consistently detect the expected abundance

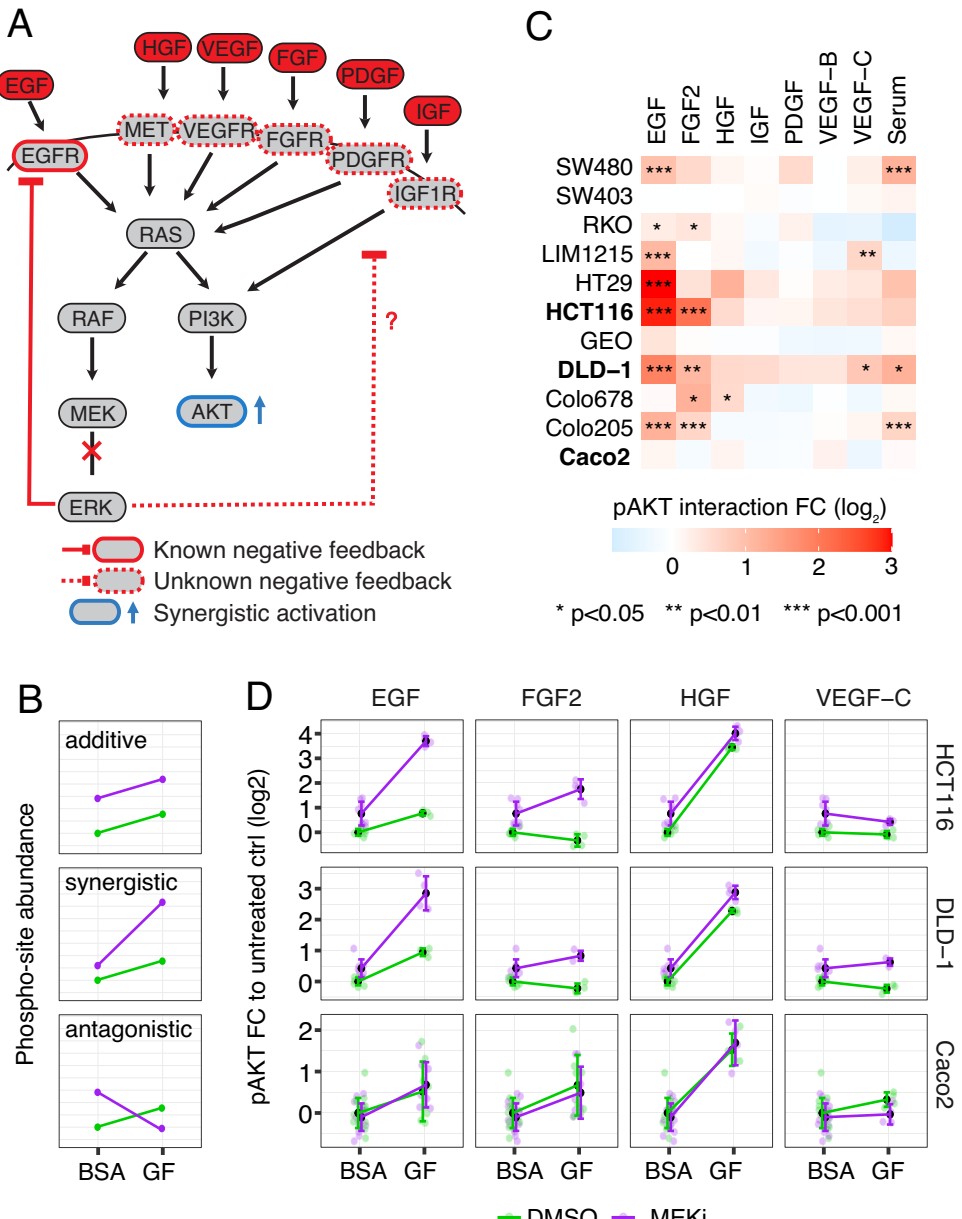

**Fig. 2 | Overview analysis CRC cell-line panel treated with MEKi and growth factor mix. A** Conceptual illustration of synergistic activation of AKT upon MEKi through receptor feedback loops and **B** Schematic explanation of interaction types. Where possible, color coding was consistent throughout the manuscript. **C** Heatmap summarizing screening results for pAKT synergistic activation across CRC cell lines upon 3.66 h MEK inhibition (1 μM Selumetinib) and subsequent 20 min stimulation with the indicated growth factors/serum, $n \geq 4$, two-way anova. **D** Quantitative Luminex measurement of pAKT response to growth factors in CRC cell lines, data are shown as mean + standard deviation, $n \geq 4$. Source data are provided as a Source Data file.

changes across the dilution series, including the negative control. As expected, quantification was better for more intense precursors. We also compared our automated DIA-NN-based workflow to semiautomatic analysis using Skyline (see Methods). Globally, the Skyline-based analysis identified fewer peptides with overall poorer quantitative performance (Supplementary Fig. 4), even though chromatographic traces for selected peptides looked convincing (Supplementary Fig. 5). We conclude that SPIED-DIA increases identification of target peptides, maintaining good accuracy and precision.

**A screen for synergistic signaling responses in 11 CRC cell lines**
Intrinsic resistance of cancer cells to targeted therapies is a significant therapeutic challenge. One known mechanism involves negative feedback from signaling pathways to upstream receptors. Inhibiting a pathway can disrupt this feedback, causing (hyper-)activation of

upstream receptors that subsequently activate parallel signaling pathways. In colorectal cancer, we and others observed that inhibiting Mitogen-Activated Protein Kinase Kinase (MEK) rapidly activates the epidermal growth factor receptor (EGFR), which synergistically enhances AKT signaling[6,7].

To more systematically investigate CRC cell responses to MEK inhibition, we initially conducted a screen across a panel of 11 cell lines. We stimulated cells with seven different growth factors or serum, both with and without the MEK inhibitor (MEKi) Selumetinib (AZD6244), and monitored AKT activation using specific antibodies (Fig. 2A). In this experiment, synergistic signaling was defined as an increase in AKT activation caused by growth factors when MEK was inhibited, meaning that the MEKi enhances growth factor-induced AKT activation (Fig. 2B). In addition to confirming the known synergy with EGF, we found that MEK inhibition also led to synergistic AKT activation via

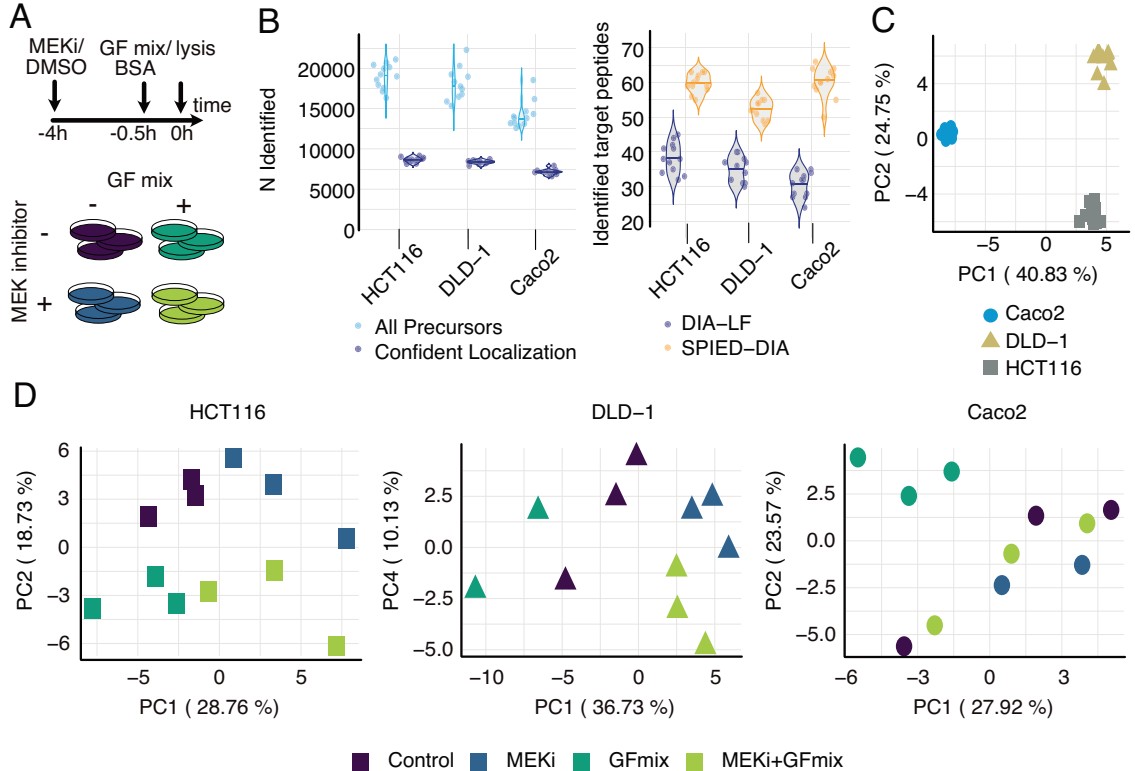

**Fig. 3 | Overview analysis CRC cell-line panel treated with MEKi and growth factor mix. A** Experimental design of CRC panel treated with MEKi (10 μM Selumetinib) and growth factor mix (GFmix, EGF, HGF, FGF2, and VEGF-C). **B** Number of identified phosphopeptides per sample. Separated according to all precursors and precursors with confidently localised phosphosites (left panel) and a comparison of identified target peptides between the label-free quantification pipeline and SPIED (right panel). **C** Principal Component Analysis (PCA) of CRC cell lines, based on precursors identified in all samples in the label free results (*n* = 1699). **D** PCA for HCT116, DLD-1, Caco2 based on precursors identified in all runs per cell line in label-free analysis: 3423, 3476, and 3185 phosphopeptides, respectively. Each point represents a sample. Variance explained by principal components is indicated. Source data are provided as a Source Data file.

FGF2 and VEGF-C in a subset of cells (Fig. 2C, Supplementary Fig. 6). Notably, the two mutant KRAS cell lines, HCT116 and DLD-1, exhibited marked synergistic responses while the KRAS wild-type cell line Caco2 did not display a notable increase in AKT activation in response to any of the growth factors (Fig. 2C, D). We therefore selected these three cell lines for more detailed analysis via SPIED-DIA.

### Targeted phosphoproteomics of synergistic signaling responses in HCT116, DLD-1 and Caco2 CRC cells

To uncover additional pathways that could contribute to intrinsic resistance through synergistic activation in CRC, we focused on HCT116, DLD-1, and Caco2 cells for in-depth phosphoproteomic analysis via SPIED-DIA. Cell lines were treated with either MEKi or DMSO control for 3.5 h, followed by 30 min exposure to a growth factor cocktail (EGF, HGF, FGF2, and VEGF-C) or BSA control (Fig. 3A). Cells were subsequently harvested and analyzed using our established workflow. In the global phosphoproteome data we identified 6000 to 8,000 confidently localised phosphorylation sites per sample (Fig. 3B). SPIED led to a two to threefold increase in the number of identified target phosphopeptides, surpassing the results from a parallel label-free analysis (Fig. 3B).

Principal Component Analysis (PCA) of the global phosphoproteomic data across all cell lines showed that cell line identity was the primary factor driving differences (Fig. 3C), confirming prior research[8]. We therefore conducted PCA on each individual cell line (Fig. 3D) to evaluate the impact of treatment. Interestingly, responses reflected the mutational profile of the cell lines. For instance, HCT116, characterized by oncogenic mutations including KRAS[G13D], PIK3CA[H1047R], CTNNB1[S45del][40,41], exhibited a noticeable shift in its phosphoproteome along the first principal component (PC1) following MEKi treatment,

with the second principal component (PC2) showing the response to the growth factor mix. Similarly, DLD-1, which has a mutational profile that includes KRAS[G13D], PIK3CA[E545K] and PIK3CA[D549N] mutations and APC truncation[42], displayed treatment responses that clustered closely together. Caco2, lacking mutations in RAS/RAF/PIK3CA, showed no phosphoproteomic shift with MEKi treatment alone. However, adding the growth factor mix caused a significant change that was largely mitigated when combined with MEKi treatment.

To further understand signaling responses, we initially focused on targeted phosphopeptides. Hierarchical clustering of differentially abundant phosphosites confirmed the expected decrease in ERK1 Tyr204 and ERK2 Tyr187 phosphorylation with MEKi treatment, alongside an increase with growth factor treatment across all cell lines (Fig. 4A–C, Supplementary Fig. 7). Phosphorylation of EGFR Tyr1172 consistently increased across all cell lines in response to growth factor treatment, confirming established signaling patterns. Additionally, we observed a potentially synergistic increase in JNK1 phosphorylation at Tyr185 in both HCT116 and DLD-1 cells, a site essential for JNK activation[43]. We note that the same phosphorylation site also maps to JNK3 (Tyr223), however this isoform is mostly expressed in the brain, heart and testes[44].

To more formally explore the synergistic regulation of targeted phosphorylation sites, we performed a factor analysis (Supplementary Fig. 8 and Material and Methods). Briefly, we defined contrasts to evaluate the individual and combined effects of growth factors and MEKi, including a special interaction contrast to detect synergies. A linear model with an empirical Bayesian estimate of variances and moderated t-test statistics was used to precisely assess the effects. This approach allowed us to accurately assess the influence of MEKi and growth factor stimulation on the levels of specific phosphopeptides,

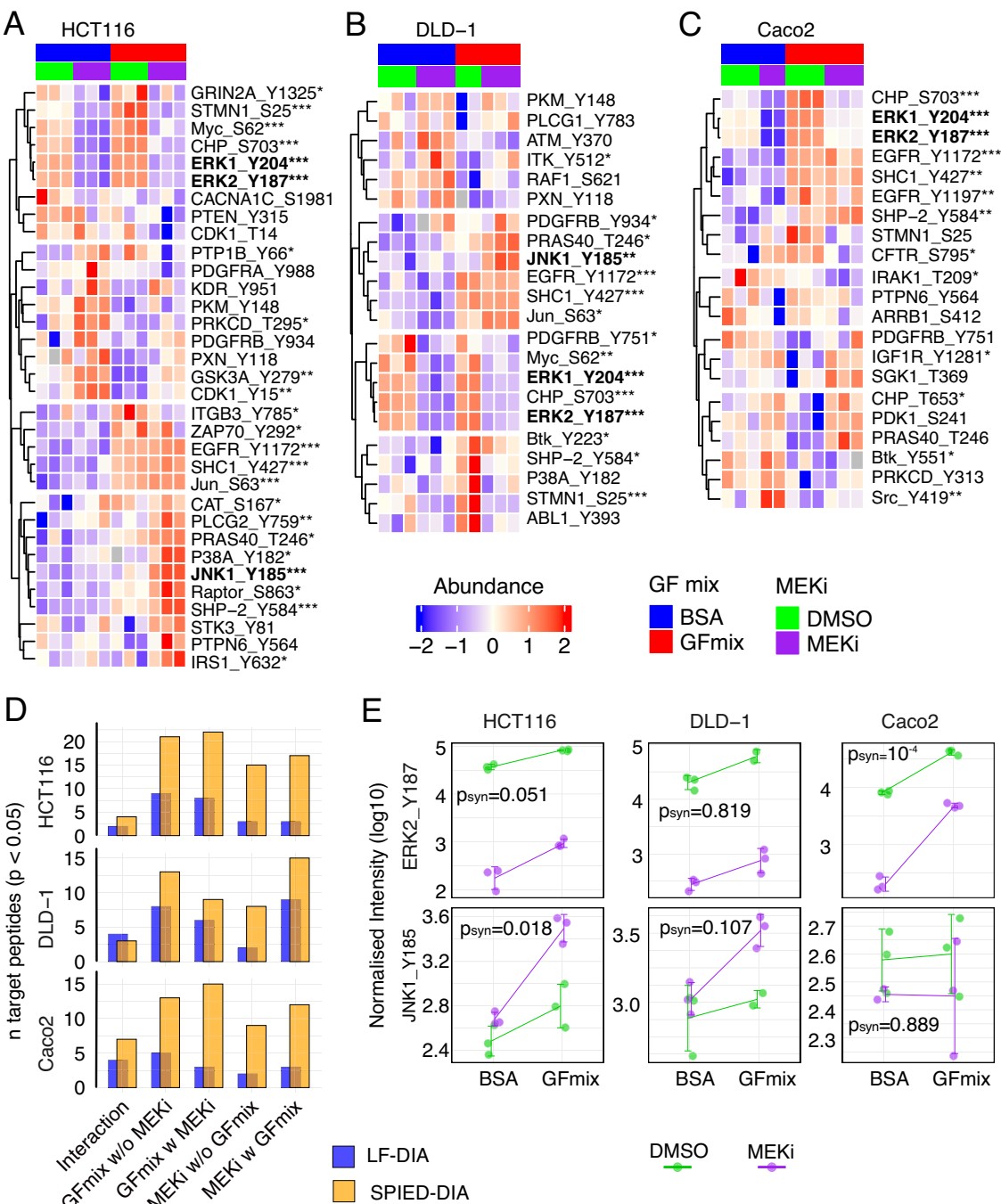

**Fig. 4 | SPIED-DIA analysis of targeted regulated phosphorylation sites.** Heatmaps display significant phosphosite regulation across different conditions in HCT116 (**A**), DLD-1 (**B**), and Caco2 (**C**) cells. All depicted phosphosites passed limma moderated F-test (two-sided) p-value cut-off of 0.1. F-test p-values for specific sites are indicated by asterisks (*, p < 0.05; **, p < 0.01; ***, p < 0.001). Color scale represents row-wise Z-normalized abundance. **D** Significantly regulated target phosphosites identified through limma moderated t-test (two-sided) p-value < 0.05. **E** Log10 transformed loess-normalised precursor-level intensities of selected phosphorylation sites. Error bars indicate standard deviation and mean and p-values derived from synergistic effect test with limma moderated t-test (two-sided). Three biological replicates unless otherwise indicated. Source data are provided as a Source Data file.

both separately and together. SPIED once again identified a larger number of regulated phosphosites, demonstrating its effectiveness (Fig. 4D). This analysis revealed synergistic regulation of a number of phosphorylation sites (Supplementary Figs. 8 and 9). JNK was indeed synergistically activated in HCT116 and to a lesser extent in DLD-1 cells (Fig. 4E). In summary, our data indicates that MEK inhibition potentiates growth factor-induced JNK activation in HCT116 and DLD-1 cells. Also, the data underscores that focusing on a small number of functionally relevant phosphorylation sites yields clear, interpretable data.

## Global phosphoproteomics confirm synergistic activation of JNK signaling in HCT116 cells

A key advantage of SPIED-DIA is its ability to yield global phosphoproteomic data alongside the targeted analysis. We used these global data to complement the findings from our targeted study. Among the 22,326 phosphopeptide precursors passing our filtering criteria across all cell lines, 3255 were identified as significantly regulated (Supplementary Fig. 8). Hierarchical clustering of these sites produced various clusters. Notably, cluster 6 in HCT116 cells and cluster 10 in DLD-1 cells

exhibited patterns suggestive of synergistic signaling (Supplementary Fig. 10, 11).

Kinase activity can be inferred from phosphoproteomic data using computational approaches[45–48]. To identify kinases involved in synergistic signaling in the global data, we performed enrichment analysis for cluster 6 and 10 in HCT116 and DLD-1 cells, respectively, using annotated kinase substrates from PhosphoSitePlus[14] and ikip-DB[49]. This analysis indicated synergistic activation of AKT1 in both HCT116 and DLD-1, consistent with the data from our screen (Fig. 3B). Although we included several AKT1 phosphorylation sites among our target peptides, the synthesis of heavy peptides failed, which explains their absence in the targeted data (Supplementary Data 1, 2). Importantly, we also observed a significant enrichment of JNK1/2 targets in cluster 6 of HCT116 cells, supporting the results from the targeted analysis (Fig. 5A). This cluster contains the well known JNK target Jun Ser63 (Supplementary Fig. 10). Hence, enrichment analysis of synergistic clusters identified kinase signatures consistent with the targeted data. Of note, no cluster indicating synergism was found in Caco2 cells, consistent with the lack of synergistic signaling responses in this cell line (Supplementary Fig. 12).

As an orthogonal way to analyze the global data, we also applied the factor analysis described above for each phosphorylation site identified in the global phosphoproteome. We then performed PTM signature enrichment analysis (PTM-SEA)[48] on phosphorylation site lists ranked by their fold-change signed p-values from the moderated t-test. Consistent with the targeted analysis results, this revealed kinase activities aligned with established biological mechanisms. For example, the growth factor induced ERK1 and ERK2 signatures ("GFmix w MEKi" and "GFmix w/o MEKi") while MEKi reduced ERK1/2 signaling ("MEKi w GFmix" and "MEKi w/o GFmix"). We confirmed synergistic AKT activation in both HCT116 and DLD-1 cells, while JNK was only significantly activated in HCT116 (Fig. 5B, Supplementary Fig. 13). In accordance with the SPIED-DIA data, no synergistic activation of JNK or AKT signaling was found in Caco2, based on the global data (Fig. 5B, Supplementary Figs. 12 and 13).

A recent study employed synthetic peptide libraries and in vitro kinase assays to systematically profile the substrate specificities of the human serine/threonine kinome, generating a comprehensive atlas that enables the prediction of kinase-substrate relationships[50]. Leveraging this independent dataset, we analyzed the enrichment of kinase motifs in the factor analysis results of our global phosphoproteomic data. This analysis revealed significant enrichment of JNK motifs in HCT116 cells, confirming a signature of synergistic JNK activation in the global dataset (Fig. 5C, Supplementary Fig. 14).

The experiments described thus far were conducted using a cocktail of growth factors. To investigate the specific contributions of individual growth factors to synergistic JNK activation in HCT116 cells, we performed additional stimulation experiments using single growth factors. For this purpose, cells were pretreated with a MEK inhibitor and then stimulated individually with EGF, HGF, FGF2, or VEGF-C, or solvent control BSA (Supplementary Fig. 15). Both targeted and global SPIED-DIA analysis (Supplementary Figs. 15 and 16) revealed that EGF, HGF, and FGF2− but not VEGF-C−are capable of mediating synergistic JNK activation. This finding suggests that the observed signaling response is broad and does not depend on a single growth factor.

In summary, the targeted and global data highlighted the same pathways. We therefore proceeded with experimental validations, measuring cell growth in HCT116 and DLD-1 cells under specific targeted treatment combinations.

### JNK inhibition sensitizes CRC cells to MEKi
Our findings revealed the involvement of both JNK and PI3K/AKT/MTOR pathways in the synergistic signaling responses to MEK inhibition. This indicates that combining MEKi treatment with drugs targeting JNK or PI3K/AKT/MTOR could be a promising strategy for

combinatorial therapy. To assess this experimentally, we treated HCT116 and DLD-1 cells with MEKi (Selumetinib) and either a JNK inhibitor (JNK-in-VIII) or a PI3K inhibitor (GDC-0941) in different concentrations and monitored cell growth over 48 h via live cell imaging (Fig. 6A).

We observed a synergistic effect of MEKi and PI3Ki on DLD-1 and HCT116 growth (Fig. 6B, C), confirming previous reports of KRAS mutated CRC models, including HCT116 and DLD-1 cells[6,51–56]. The JNK inhibitor (JNKi) had minimal effect on DLD-1 cell division, both alone and when combined with MEKi (Fig. 6D, E). In contrast, a 5 μM dose of JNKi significantly slowed the division of HCT116 cells, nearly doubling the cell division time. MEKi alone exhibited a more modest impact, even at the highest concentration tested. Notably, while low doses of JNKi or MEKi alone did not significantly affect HCT116 cell division, their combination markedly inhibited cell growth (Fig. 6D). Specifically, 1 μM JNKi or 0.2 μM MEKi alone extended HCT116 cell cycle times from 25 to 27 h, but combining both drugs increased this to 37 h. Hence, MEK inhibition sensitizes HCT116 cells to JNKi. Collectively, these results show that our targeted phosphoproteomic approach has uncovered synergistic signaling responses that are functionally relevant.

## Discussion
In this study, we explored synergistic signaling in colorectal cancer cell lines by integrating DIA with spike-in synthetic heavy stable isotope-labeled phosphopeptides, enhancing the detection of specific phosphopeptides. Our method, SPIED-DIA, marries the sensitivity of targeted detection with DIA's broad discovery capabilities, offering a streamlined approach that does not require specialized data acquisition schemes and is compatible with standard mass spectrometers, simplifying the detection of key phosphorylation sites while leveraging the full benefits of label-free DIA phosphoproteomics. Despite these advantages, SPIED-DIA also has limitations. SPIED-DIA provides only a modest sensitivity boost for target peptides because it lacks the longer selective ion collection periods found in other targeted approaches[29,30,37]. Moreover, like other spike-in strategies, obtaining target-specific heavy reference peptides is challenging and expensive. To economize, we opted for pooled synthesis of heavy reference peptides. However, this approach led to unsuccessful synthesis of some desired peptides, rendering them unusable in our targeted strategy. A viable alternative is utilizing off-the-shelf reference peptide collections like the PQ500 standard for plasma proteomics (Biognosys), the kinase activation loop peptides collection (JPT) or the recently introduced multipathway phosphopeptide standard (Thermo)[57]. Another advantage of these standardized sets is that they also enhance data consistency across different research labs. The community would benefit if other target sets like the SigPath and T-loop libraries were to also become commercially available[26,27]. Ideally, such target sets would include peptides from MAPK activation loops that are phosphorylated at both threonine and tyrosine residues, as phosphorylation at both sites is essential for activation. Moreover, including non-phosphorylated versions of target phosphopeptides in the spike-in strategy could provide additional insights into changes in protein abundance and phosphorylation site occupancy.

In CRC, the classical MAPK signaling pathway is chronically activated, often by mutations in RAS or RAF[58]. This suggests that inhibiting the central kinase, MEK, could be a valid therapeutic strategy. However, signaling in CRCs rewires due to strong feedbacks leading to reactivation of the pathways as well as cross-activation of parallel pathways[7,59,60], often mediated by the EGF receptor. Based on a screen with 11 CRC cell lines and different growth factors, we observed that MEKi-induced synergistic AKT activation is not only mediated by EGF but also by HGF, FGF2 and VEGF-C. We then applied SPIED-DIA to investigate synergistic signaling responses in three selected cell lines in detail. Our results highlight the key advantages of the SPIED-DIA approach. Firstly, the targeted analysis of functionally relevant

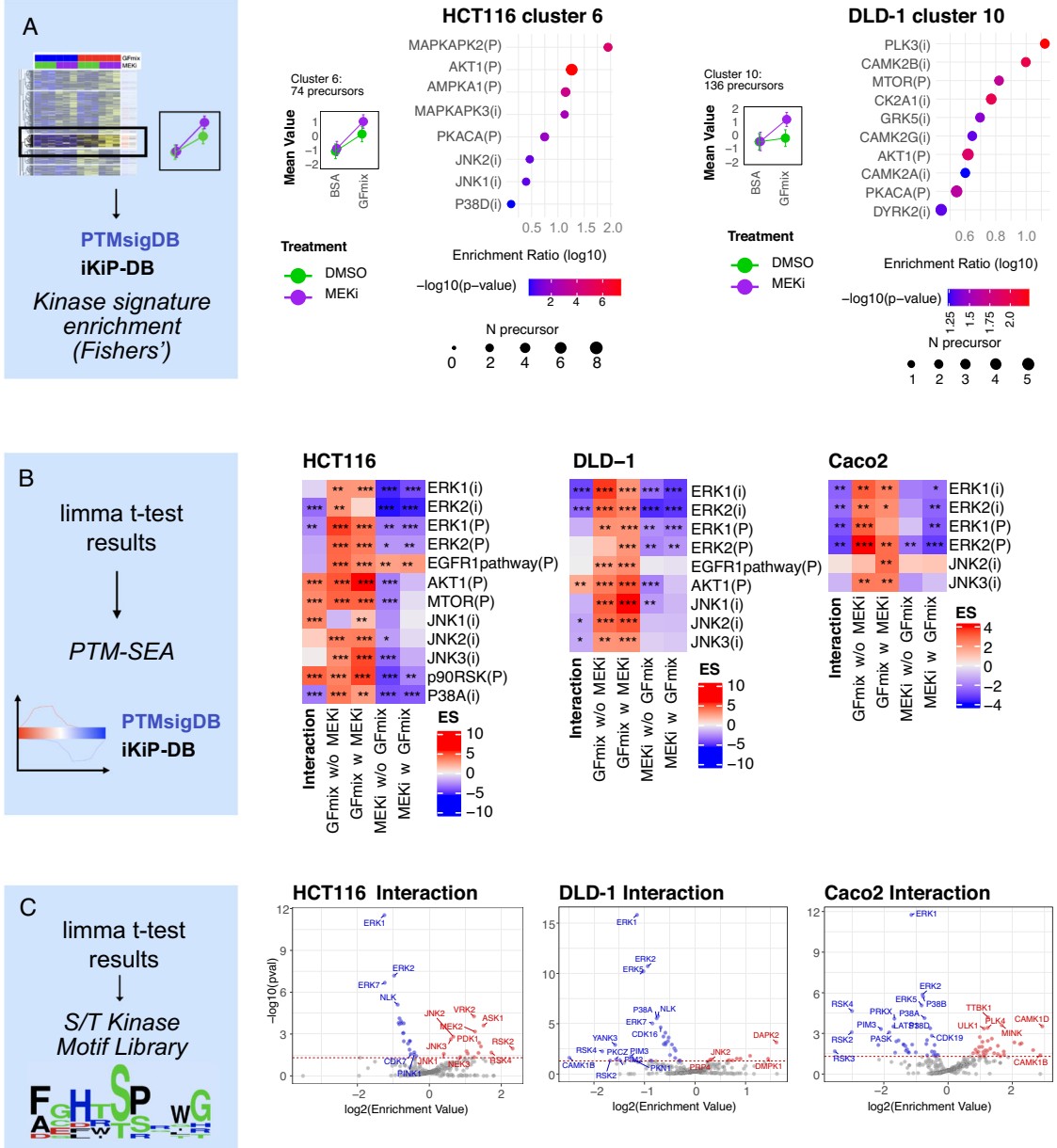

**Fig. 5 | Analysis of label-free quantification data and kinase activity profiling.**
**A** Kinase signature enrichment analysis of synergistic profile clusters from hierarchical clustered label-free data. Z-scored intensity profiles of treatment conditions within clusters. Data represented as mean ± standard deviation. Kinase signatures selected from PhosphoSitePlus and iKiP-DB. Size and color of points indicate number of precursors and significance (Fishers' exact test), respectively. Biological replicates as indicated in Fig. 4. **B** Selected results from PTM-SEA using PhosphoSitePlus (PSP) and iKiP-DB kinase signatures, denoted by (P) and (i), respectively. PTM-SEA input consists of fold change signed p-values from limma moderated t-test (two-sided), filtered for phosphopeptides with moderated F-test (two-sided) p-value < 0.1, indicating significant regulation in at least one test. ES = enrichment score as calculated by PTM-SEA. Significance is denoted by asterisks, with * p < 0.1, ** p < 0.05, *** p < 0.01. **C** Kinase Library S/T (Serine/Threonine) Kinase Motif Enrichment Analysis. To derive 'foreground' phosphosites, moderated t-test results were filtered for a fold change (FC) > 0.1 and a p-value < 0.05. The output is the enrichment value (EV). Results for interaction test are depicted. Source data are provided as a Source Data file.

phosphorylation sites provides easily interpretable data that directly highlight key changes in cell signaling. Secondly, the global data obtained in parallel can be used to extract kinase activity profiles. Interestingly, both approaches revealed synergistic activation of JNK signaling in HCT116 cells, suggesting that combinatorial treatment with MEKi and JNKi could be an attractive therapeutic option. The consistency between the datasets not only validates our analytical strategy but also solidifies our confidence in these candidate pathways as critical mediators of the cellular response in our study model. Indeed, we confirmed that JNK inhibition sensitizes HCT116 cells to MEKi treatment, leading to marked reduction in cell proliferation. In

addition to EGF, JNK was also activated by HGF and FGF2, supporting the notion that synergistic signaling can also be mediated by other receptors than EGFR.

Oncogenic signaling in CRC involves multiple pathways, notably EGFR/MAPK, WNT, Pi3K/Akt, JAK/STAT, Notch, SHH, and TGF-beta[3,61]. Given that JNK is not typically linked to oncogenic signaling in CRC, our observation of its synergistic activation upon MEK inhibition is initially surprising. However, our data does align with known MEK-JNK interactions. MEK activation typically upregulates DUSP4, which inactivates JNK[62–65]. Additionally, cancer cells with loss of function mutations in JNK activating kinases MAP3K1 or MAP2K4 are often sensitive to MEK

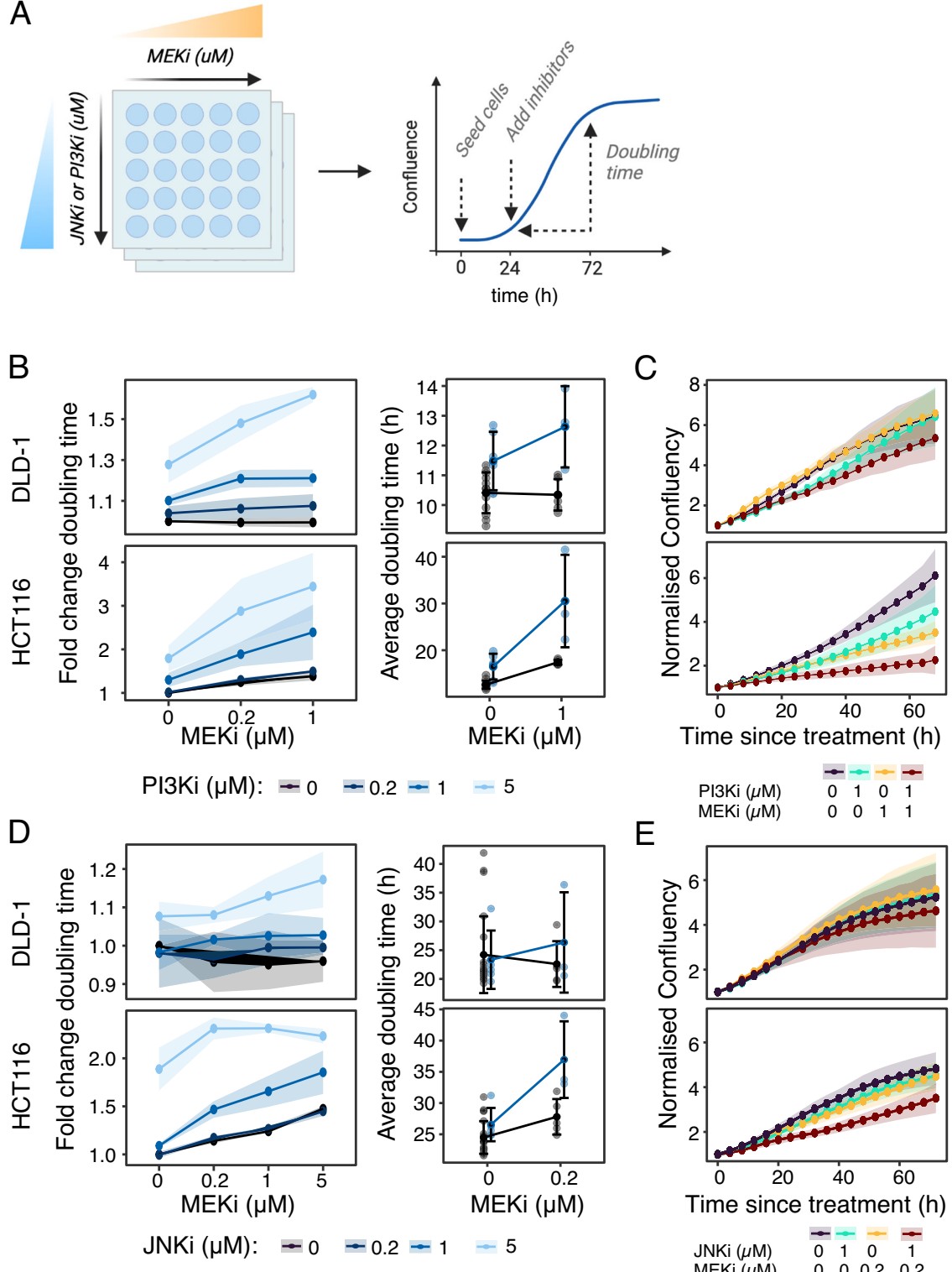

**Fig. 6 | Experimental workflow and results for combination treatment with MEKi and JNKi or PI3Ki. A** Experimental workflow depicting inhibitor treatment of cell lines prior to measurement of cell growth and doubling time with live-cell imaging. **B** Cell doubling time fold changes relative to within replicate controls, 48 h post-treatment with inhibitors. Insets depict doubling times at selected concentrations. **C** Growth curves of cells within selected treatment conditions, normalized to treatment-start. **D**, **E** Similar to panels **B**, **C**, these graphs display the effects of combining MEKi with JNKi on cell doubling time and growth curves. Data are presented as mean + standard deviation, shaded ribbons also indicate standard deviation. $n \geq 3$. Source data are provided as a Source Data file.

inhibitors, highlighting the functional relevance of the ERK-JNK crosstalk[66]. Very recently, the KRAS[G12C] inhibitor sotorasib and the MAP2K4 inhibitor HRX-0233 were shown to synergistically inhibit growth of a number KRAS mutant CRC and lung cancer cell lines and to

induce durable tumor shrinkage in mouse xenografts of human lung cancer cells[67]. Although this recent study focused on different kinases, its findings bolster the concept of synergistically targeting MEK/ERK and JNK signaling as a viable approach in cancer therapy.

Selecting optimal treatments to target cancer remains an important challenge. In this study, we demonstrate that targeted phosphoproteomics using SPIED-DIA can reveal signaling responses, which can help predict effective combinatorial treatments. While the work presented here is limited to cancer cell lines, the technology also has the potential to be applied to patient samples. For example, microscaled phosphoproteomic techniques have successfully identified 7000 phosphosites in retrospective formalin-fixed paraffin-embedded (FFPE) tissue samples[68]. Similarly, a recently developed low input phosphoproteomics workflow enabled detection of 6000 to 17,000 phosphosites from as little as 1 to 20 μg protein starting material[69]. Finally, combining ultrasensitive proteomics with laser tissue microdissection facilitates spatial proteomics in human tissues[70,71]. In the future, integrating these technologies with SPIED-DIA promises to reveal signaling pathways and predict treatments directly in patient samples.

## Methods

### Cell culture and treatment experiment
HCT116, Caco2 and DLD-1 were obtained from ATCC (Manassas, Virginia, USA). Cell lines were confirmed to be mycoplasma free. All cell lines were cultured in DMEM, high glucose, GlutaMAX Supplement, pyruvate (Gibco, Invitrogen), supplemented with 10% fetal bovine serum (FBS, Gibco), in a humidified incubator at 37 °C with 5% $CO_2$. Experiments are performed at 80% confluence. Cells are harvested in ice cold PBS, by washing two times in ice cold PBS, and scraping from plate. The cells were rinsed again with 1× PBS and centrifuged at 250 g for 5 min in a centrifuge maintained at 1 °C. The final PBS wash was removed and the resulting pellet was frozen in ethanol on dry ice and stored at −80 °C.

### Treatment with Growth Factor and MEK Inhibitor
For the interaction experiment, we used the MEK inhibitor Selumetinib (AZD6244, #S1008, Selleckchem) at a final concentration of 10 μM, dissolved in DMSO. The growth factors used were HGF (Peprotech #100-39H, final concentration 0.025 μg/ml), FGF2 (Peprotech #100-18B, final concentration 0.005 μg/ml), EGF (Peprotech #AF-100-15, final concentration 0.025 μg/ml), and VEGF-C (Peprotech #100-20CD, final concentration 0.1 μg/ml), all of which were dissolved in 0.1% BSA. Depending on the specific experiment, these growth factors were either added individually or combined into a mixture.

Prior to treatment, the cells underwent a starvation period of approximately 18 h using 0.1% FBS to synchronize their growth. Following this, the cells were treated with either MEKi or a control solution of DMSO for 3.5 h. At 3.5 h either a control solution (0.1% BSA) or the growth factor (mixture) was added to the cells for another 30 min. After a total incubation time of 4 h, the cells were harvested for further analysis.

### Selection and synthesis spike-in peptides
Phosphosites were selected for relevance to cellular signaling. The Kinase-Substrate, Phosphorylation-site and Regulatory-site datasets were downloaded from PhosphoSitePlus (February 2018). Phosphorylation sites that are present in all three datasets, annotated to regulate protein activity and annotated to selected KEGG signaling pathways. Phosphosites annotated to manually selected proteins (CREB1, ABL1, IGF1R, IRS1, RPS6KA1, PDGFRA, PIK3R1, and RPS6) or with more than 100 references were kept in regardless of activity and pathway annotation. This results in a list of ~1000 phosphorylation sites. These phosphorylation sites were mapped to an in-silico trypsin digested proteome, and the resulting phosphopeptides were filtered based on their MS properties. The peptide is not allowed to be found in the proteome >10 times and it can not contain >10 phospho-accepting residues. Phosphopeptides were filtered for synthesis feasibility, and singly phosphorylated peptides with a length between 7 and 21 amino acids and N-terminal K/R were selected for synthesis. The list with

shorter peptides contains 524 phosphorylation sites, mapping to 485 unique phosphopeptides, on 277 proteins. Peptides were purchased from JPT, synthesized using FMOC solid-phase technology with crude purity and synthetic isotope–labeled c-terminal lysine (K) or arginine (R) and pooled. These SpikeMix™ Peptide Pools are more cost-effective than synthesizing individual peptides; however, it is important to note that all peptides on the list are anticipated to be synthesized successfully with adequate yields. For each experiment, we always injected this heavy peptide mixture to create an experiment specific library. In total, 274 unique heavy phosphopeptides (108 pTyr, 43 pThr, 123 pSer peptides) were detected in at least one experiment. The list of the initially selected peptides and detected peptides (in library) can be found in Supplementary Data 1. Lyophilized synthetic peptide pools were kept at -20.

### Phosphoproteomics sample preparation
Cell pellets were lysed at 4 °C with urea lysis buffer (8 M urea, 50 mM Tris (pH 8),150 mM NaCl) supplemented with protease inhibitors (2 μg/ml aprotinin, 10 μg/ml leupeptin) and phosphatase inhibitors (10 mM NaF, phosphatase inhibitor cocktail 1 and 2, Sigma Aldrich). The cell lysate was treated with 5 mM dithiothreitol for 1 h to reduce proteins and then alkylated with 10 mM iodoacetamide for 45 min in the dark. Sequencing grade LysC (Wako) was added at a weight to weight ratio of 1:50. After 2 h, samples were diluted 1:4 with 50 mM Tris–HCl pH 8 and sequencing grade trypsin (Promega) was added at 1:50 ratio. Digestion was completed overnight. subsequently samples were acidified using FA and desalted with Sep-Pak C18 cc Cartridges (Waters). Lyophilized samples are diluted to 0.7 μg/μL in 80% ACN/0.1% TFA. 200 fm heavy labeled synthetic peptides are added to 100 μg sample and subjected to automated immobilized metal affinity chromatography (IMAC) phosphopeptide enrichment by the Bravo Automated Liquid Handling Platform (Agilent) with AssayMAP Fe(III)-NTA cartridges[72].

### Liquid chromatography mass spectrometry
Mass spectrometry raw data were acquired on a Bruker timsTOF Pro2 connected to a Thermo Fischer EASY-nLC 1200 system. Around 300 ng (1/3 of IMAC output) was injected. Samples were separated online on a 25 cm column packed in-house with C18-AQ 1.9 μm beads (Dr. Maisch Reprosil-Pur 120). A gradient of mobile phase A (0.1% formic acid and 3% acetonitrile in water) and mobile phase B (0.1% formic acid, 90% acetonitrile in water) was used to separate the peptides at a flow rate of 250 nl/min. Mobile phase B was ramped from 2% to 30% in the first 29 min, followed by an increase to 60% B in 3 min and a plateau of 90% B for 5 min. Temperature of the column was kept constant at 45 °C. The LC system was connected to Bruker timsTOF Pro2 hybrid TIMSQTOF mass spectrometer via a CaptiveSpray nano-electrospray source. The raw files were acquired in dia-PASEF mode, using the standard 'long gradient' method as supplied by the vendor. All spectra within a mass range of 400 to 1201 Da and an IM range from 1.6 to 0.6 V s/cm$^2$ were acquired using equal ion accumulation and ramp times in the dual TIMS analyzer of 100 ms each. The collision energy was lowered as a function of increasing ion mobility from 59 eV at 1/K0 = 1.6 V s/cm$^2$ to 20 eV at 1/K0 = 0.6 V s/cm$^2$. The estimated cycle time is 1.80 s. The calibration status of the machine is monitored constantly and calibration of the ion mobility dimension is performed linearly using at least three ions from Agilent ESI LC/MS tuning mix (m/z, 1/K0: 622.0289, 0.9848 V s/cm$^2$; 922.0097, 1.1895 V s/cm$^2$; 1221.9906, 1.3820 V s/cm$^2$).

### Benchmark sample generation and High-pH Reverse-Phase Fractionation for Library Generation
For the High pH library generation HCT116 was treated with combinations of phosphatase inhibitors, to increase the number detectable phospho-site relative to normal growth conditions. For inhibition of phosphatases, HCT116 (ATCC, #7) was treated with 1 mM pervanadate and 50 ng/ml calyculin A. To this end, cells were starved in 0.1% FBS for

3 h. afterwards cells were treated with no serum, with 10% FBS, 10% FBS + calyculin A, or 10% FBS + calyculin A + Pervanadate. Samples were processed as described before, and desalted peptides were combined before drying down.

For library generation, the peptides are subjected to offline high pH reverse phase fractionation by HPLC on an Agilent 1290 Infinity II HPLC instrument. To this end, the dried peptides were reconstituted in high pH buffer A (4.5 mM ammonium formate, 2% ACN, pH 10), and loaded on a XBridge BEH C18 4.6 × 250 mm column (130 Å, 3.5 μm bead size; Waters), and separated using a 96-min gradient with a flow rate of 1 ml/min. The gradient was performed by ramping high pH buffer B (4.5 mM ammonium formate, 90% ACN, pH 10) from 0% to 60%[72]. The 96 fractions were collected and concatenated by pooling equal interval fractions. The final 48 fractions were dried down and resuspended for IMAC enrichment as described above. A total of 100 μg of each pooled fraction were used for IMAC enrichment. Per IMAC enriched fraction 100 ng phospho-enriched peptides were measured on the timsTOF Pro2.

Same LC conditions as described previously were used. Data were acquired using default DDA-PASEF mode with a cycle time 1.1 s and 10 PASEF MS/MS scans per topN acquisition cycle. All spectra were acquired within an m/z range from 100 to 1700 and an IM range from 1.6 to 0.6 V s/cm$^2$.

Raw data was analyzed with MaxQuant (v2.4.0.0) and searched against the human reference proteome database (downloaded from UniProt in 06/2023) and default protein contaminants included in MaxQuant. Fixed modifications were set to carbamidomethylation of C. Variable modifications included oxidation (M) and N-terminal acetylation and phosphorylation (STY). A maximum of 5 modifications per peptide and 2 missed cleavages were allowed. MaxQuant results are filtered to exclude reverse database hits, potential contaminants and phospho-sites with a localisation probability lower than 50%. MaxQuant results were transformed and were necessary combined to a DIAN-NN compatible library, including ion mobility information.

### Synthetic peptides library generation

The synthetic phosphopeptides were measured to generate a library. The peptides were dissolved in Buffer A (3% ACN, 0.1% FA). To generate a library, 50 fm and 100 fm peptides were measured in DDA-PASEF mode in triplicates, with the same settings as for HpH-library generation acquisition. The resulting raw files were analysed in MaxQuant against a library specific.fasta file. MaxQuant settings and processing of MaxQuant output as above. A table with peptides in the library, along with main annotated phosphosite and other annotated phosphosites can be found in supplementary data 1.

### Full phospho-proteome dilution benchmark

For dilution series with the SILAC labeled phosphoproteome HCT116 was cultured in Heavy or Light SILAC medium. SILAC medium consists out of arginine- and lysine- free DMEM, supplemented with 10% dialyzed fetal bovine serum (dFBS, Gibco) and either heavy (13C615N4 L-arginine or Arg10 an 13C615N2 L-lysine or Lys8) and light amino acids (Cambridge Isotope Laboratories) at 0.4 mmol/L and 0.8 mmol/L. Cell harvest and lysis as described above. Protein concentration in cell lysate was determined using BCA and the heavy labeled cell lysate was sequentially diluted into the light cell lysate. Subsequent sample preparation and phosphopeptide enrichment were performed as usual. For mass spectrometry 100 ng phospho-enriched peptides per sample were injected per dilution in triplicates and analysed in DIA-PASEF mode, as described above.

### Cell line panel screen for MEKi-dependent receptor-mediated feedbacks

**Bio-Plex data generation.** Human colorectal cell-lines used in this experiment Colo205, Colo678, DLD-1, GEO, HCT116, HT29, LIM1215,

RKO, SW403, SW480 and Caco2 were provided by AG Sers Molekulare Tumorpathologie (Charité-Universitätsmedizin). All cell-lines were cultured in low glucose DMEM (D5546-6X500ML, Sigma-Aldrich) supplemented with 10% FBS, 10 mM Ultraglutamine and Penicilin-Streptomycin and were incubated at 37 °C and 5% CO₂. Before perturbation commenced cells were starved overnight in serum free medium. At 4 h before lysis the cells were treated with 1 μM AZD6244 (Selleckchem, S1008) or solvent control DMSO and at 20 min before lysis cells were stimulated with ligands, full serum (10% FBS) or solvent control PBS/BSA ($n = 4$ replicates). We used the following ligands (all Peprotech): EGF (25 ng/ml), HGF (50 ng/ml), IGF1 (100 ng/ml), FGF2 (5 ng/ml), PDGF (10 ng/ml), VEGF-B (100 ng/ml) and VEGF-C (100 ng/ml). After treatment and incubation, lysates were collected and analyzed with the Bio-Plex Protein Array system (Bio-Rad, Hercules, CA) as described earlier using magnetic beads specific for AKT$^{S473}$, ERK1/2$^{T202,Y204/T185,Y187}$ and MEK1$^{S217,S221}$. The beads and detection antibodies were diluted 1:3. For data acquisition, the Bio-Plex Manager software and the R package lxb was used.

**Bio-Plex data processing.** First, obvious outliers among replicates exhibiting an absolute z-score >=3 for all three phosphosite measurements were removed. For each cell-line, data were processed separately for each of the 3 measured phosphoproteins. The value of the control (PBS/BSA + DMSO) was estimated as the mean value of the replicates and log2 fold changes with respect to the control were then computed for all conditions. The resulting fold changes $x$ where then used to calculate the hyperactivation effect of GF and AZD on pAKT:

$$AKT_{interaction\_FC} = \mu(x_{GF+AZD}) - \mu(x_{GF+DMSO}) - \mu(x_{BSA+AZD}) - \mu(x_{BSA+DMSO}).$$

To estimate the significance of this hyperactivation we conducted a two-way anova analysis with interaction term and ascribed synergistic hyperactivation if meeting the following three criteria: (i) significance (p < =0.05), (ii) synergy ($AKT_{interaction\_FC} > 0$) and (iii) receptor dependency ($AKT_{interaction\_FC}(GF) > AKT_{interaction\_FC}(PBS/BSA)$ in the same cell line).

### SPIED-DIA analysis

**Raw file processing.** For the label-free analysis of the raw files, the raw files were processed using DIA-NN (v1.8.2 beta 11), with searches conducted against the library derived from the target peptides only (generated as described above) and reannotation enabled. Settings included methionine excision and in silico digestion at K/R, with cysteine carbamidomethylation as a fixed modification. Variable modifications included methionine oxidation, N-terminal acetylation, and phosphorylation on STY, with phosphorylation scored independently. The analysis allowed for one missed cleavage and a maximum of three variable modifications. The "report-lib-info" option was activated to facilitate raw data verification in subsequent stages. SILAC labeling with a mass delta of 0 at KR was applied as a fixed modification, SILAC channels L (K[0], R[0]), H (K[8.0142], R[10.0083]), and a decoy (K[16.0284], R[20.0165]) were registered.

**Process DIA-NN output.** Data are filtered to only include Heavy channel entries (spike-in), Channel.Q.Value < 0.05, PTM.Q.Value < 0.05, PTM.Site.Confidence > 0.5 and a Channel.H > 1000 (spike-in intensity) and identified in >9 samples (see supplementary table 2 for an overview of consistently identified heavy peptides). Furthermore, precursors need to have Channel.L (light, endogenous intensity) > 900 in at least 1 condition to ensure no noise-to-noise comparisons and the light precursor needs to be identified with a Channel.Q.Value < 0.5 in at least 3/12 samples. Subsequently, for the precursors passing the filters, a "rescalingfactor" is calculated using the median of Channel H intensities, and light intensities are rescaled to log10-transformed ratios of Channel L to Channel H, adjusted by this factor like such: log10((-Channel.L/Channel.H)*rescalingfactor). This transformation is applied to mitigate intensity disparities and inspired by the RefQuant approach[32].

The rescaled intensities are normalized using the normaliseCyclicLoess function from the limma package. The differential abundance analysis is performed as described in the label-free data analysis pipeline. Precursors are grouped by unique phosphopeptide sequence and filtered for precursors with the lowest F-test p value. Precursors with an F test p value < 0.1 are selected for visualization in a heatmap.

**Visualization raw data.** For the visualization of the raw MS/MS spectra, which facilitates validation of phosphorylation site localisation and the identification of stable isotope-labeled fragments, we employed the following procedure: the ScanID was retrieved from the DIA-NN output table. To determine the corresponding exact scan number from the Bruker raw file, we treated the approximate scan number as the absolute number of MS/MS scans within the run. The exact scan number was then directly derived from the raw data file itself. Subsequently, the MS/MS spectra were downloaded using the Bruker Data Analysis tool. Relevant peaks within the spectra were manually annotated in R using the spectral library as used for the DIA-NN analysis within a 10 ppm mass accuracy range and plotted.

## Label-free DIA analysis

**Raw file processing.** For the label-free analysis of the raw files, the raw files were processed using DIA-NN (v1.8.2 beta 11), with searches conducted against the high pH library (generated as described above) and reannotation enabled. Settings included methionine excision and in silico digestion at K/R, with cysteine carbamidomethylation as a fixed modification. Variable modifications included methionine oxidation, N-terminal acetylation, and phosphorylation on STY, with phosphorylation scored independently. The analysis allowed for one missed cleavage and a maximum of three variable modifications. The "report-lib-info" option was activated to facilitate raw data verification in subsequent stages.

**Filter and normalise DIA-NN output.** DIA-NN output was processed in R (v4.3.0) filtered with Q.Value < 0.05, only phosphorylated precursors, PTM.Site.Confidence > 0.5 and PTM.Q.Value < 0.05 (Supplementary Fig. 1). Precursor intensities (Ms1.Area) were log10-transformed and collectively normalised to correct for loading bias between samples using loess (function: normalizeCyclicLoess) from the limma package (v3.56.1)[73]. No imputation was performed at any stage in the analysis. PCA was performed on precursors identified in every sample within a group (all cell lines together or individual cell lines).

**Differential abundance analysis.** Differential abundance analysis of phosphopeptides within cell lines, across conditions, was conducted using the limma package, employing a factorial analysis approach with MEKi and GFmix as factors in the linear model. Precursors were filtered to include only those with a maximum of five missing values. Within the factorial design, contrasts were strategically defined to investigate synergistic effects: the differential impact of the growth factor mix with and without MEKi ("GFmix w MEKi" and "GFmix w/o MEKi"), and conversely, the effect of MEKi with and without the growth factor mix ("MEKi w GFmix" and "MEKi w/o GFmix"). Potential synergistic interactions were explored through an "Interaction" contrast. A linear model was fitted to the data and Bayesian statistics (*ebayes* function limma) were then applied to estimate variance among the precursors, employing moderated t-statistics (two-sided) and moderated F-statistics (two-sided). Results were extracted and aggregated for further analysis. Especially, precursors are grouped per unique phosphopeptide sequence and the precursor with lowest p-value from the moderated F-test is selected for downstream analysis.

**Clustering significantly interesting phosphosites and investigation interesting clusters.** Precursors with F p-values < 0.1 (as indicated below figure), indicating significant regulation, were selected and displayed in a heatmap. Hierarchical clustering was used to organize the heatmap, with the number of clusters determined manually to best represent the data. For each cluster, means of z-score normalized precursors were calculated. Clusters suggesting synergistic interactions between GFmix and MEKi were specifically identified for further analysis. Kinase signatures from PTMsigDB and iKIP-DB were used to perform kinase overrepresentation analysis via Fisher's exact test, identifying enriched kinase activities linked to the treatment effects.

**PTM-SEA analysis.** PTM signature enrichment analysis (PTM-SEA, https://github.com/broadinstitute/ssGSEA2.0) was employed to infer kinase activity from regulated phospho-sites. Precursors with a p-value lower than 0.1 as derived from the limma moderated F-statistics (regulated phosphopeptides) were selected. As input we used signed (according to log2 fold change) -log10-transformed p-values per comparison, derived from moderated t-test. PTM signatures were sourced from PTMsigDB[48] (v2.0.0) and iKIP-db[49]. As unique site identifiers, the 14 amino acid phospho-site flanking sequence window was used. Multiply phosphorylated peptides were split per phosphorylation site. PTM-SEA was run with sample.norm.type set to "none" and weight to "1".

**S/T kinase motif enrichment analysis.** For S/T kinase motif enrichment analysis, we utilized the S/T kinase library tool (https://kinase-library.phosphosite.org/), as detailed in its foundational paper[50]. The input comprised the precursor amino acid sequences, in accordance with the tool's specifications. Data input included fold changes and p-values for each comparison, obtained from the moderated t-test. We applied a fold change threshold of greater than 0.1 and a p-value threshold of lower than 0.05 to select for significantly abundant phospho-sites. All predicted kinases were represented in volcano plots, highlighting those significantly associated with the observed phosphopeptide regulation.

## Skyline analysis

For alternative validation of SPIED-DIA the dilution experiment has been analysed in addition with Skyline (24.1.0.199). Following adjustments in the settings have been applied. Peptide Settings: Structural Modifications: Phospho (STY), Ammonia Loss (KNQR), Water Loss (D,E,S,T), Acetyl (n-term),Oxidation (M);Isotope Modifications: 13 C(6) 15 N(4) (R), 13 C(6)15 N(2) (K); Internal Standard Type: heavy: Precursor Charges: 1,2,3,4, IonTypes: y,b,a Retention time filtering: Only scans within 10 min of MS/MS IDs. The heavy library was imported using the 'insert transition list' option. Peptides were inserted with their corresponding known modifications using the 'insert Peptides' option. The report feature was used to export Transition Results tables including precursor intensity information. In addition chromatograms were exported and corresponding peaks as identified by Skyline have been plotted for selected peptides using R and the ggplot2 package. Light to heavy ratios were calculated from fragment areas.

## Growth curves inhibitor combination treatment HCT116 and DLD-1

**Combination Treatment.** The effects of combination treatment were assessed by monitoring cell proliferation and death through live-cell imaging. In validation experiments, HCT116 and DLD-1 cell lines were treated with combinations of a MEK inhibitor (AZD6244, Selumitinib) and either a JNK inhibitor (JNK-IN-8, S4901, Selleckchem) or a PI3K inhibitor (Pictilisib, GDC-0941, S1065, Selleckchem). Cells were seeded at densities of 4000 cells per well for HCT116 and 2500 cells per well for DLD-1 in 96-well plates and cultured in the described growth medium. Twenty-four hours post-seeding, cells were treated with inhibitor combinations at concentrations of 0 (duplicated for double-negative controls), 0.2, 1, and 5 µM, using a quadratic mixing

format. To mitigate edge effects, outer rows were left empty and filled with PBS. Cell growth was monitored for an additional three days post-treatment. Experiments were performed in biological triplicates. For treatments combining MEK and JNK inhibitors, the protocol included an additional condition where the medium was supplemented with growth factors (HGF, EGF, and FGF) at specified concentrations.

**Incucyte Live Cell Imaging.** Automated phase-contrast and green-fluorescent long-term imaging was conducted using an Incucyte instrument (dual-color model 4459, Incucyte Essen Bioscience) in a standard humidified incubator at 37 °C and 5% $CO_2$. Imaging occurred every four hours, capturing four frames per well using a Nikon 10x objective.

**Image Processing.** Images were processed using Incucyte ZOOM software (2018A) with the manufacturer's default masking settings. Confluence values (percentage of area covered by the confluence mask) were exported for further analysis. Image frame data were individually exported and processed in R.

Growth was assessed in multiple ways: raw growth curves were examined for outliers and excluded from further analysis. To determine changes in doubling time, the doubling time in the 48 h post-treatment was compared to the baseline (0 μM concentration) within each replicate. The average doubling time for selected concentrations of interest was summarized across all replicates. The growth curves for selected concentrations were normalized to the confluence at time zero (treatment) within each well or image frame.

### Reporting summary
Further information on research design is available in the Nature Portfolio Reporting Summary linked to this article.

## Data availability
Mass spectrometry raw files as well as MaxQuant and DIA-NN output files, spectral libraries and associated files have been deposited to the ProteomeXchange Consortium via the PRIDE[74] partner repository. The accession ID is PXD050961. The processed data such as normalized intensities and results of the differential expression analysis as derived from the proteomics experiments are available as Supplementary Data files. The raw data of the Bio-Plex and live-cell imaging experiments are provided as Supplementary Data files. Source data is provided for Figs. 1D, 1E, 1F, 1I, 2C, 2D, 3C, 3D (Caco2, DLD-1, and HCT116), 4A, 4B, 4C, 4E, 5C (plus Supplementary Fig. 14), 6B (left/right), 6C, 6D (left/right), 6E, and the following Supplementary Figs. 1A, 1B, 1C, 3, 4, 9, and 15C. Source data are provided with this paper.

## Code availability
The R scripts necessary to perform SPIED-DIA analysis, as well as the scripts to perform the analyses relevant to the main figures in the manuscript can be found at https://github.com/Mirjamva/SPIED-DIA. The repository was linked to Zenodo (https://doi.org/10.5281/zenodo.15045498)[75].

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

## Acknowledgements

We would like to thank Martha Hergeselle (MDC) for her help with cell culture and Christian Sommer (MDC) for technical support. We also thank Vadim Demichev (Charité), Anna Welter and Robert Kerridge (all MDC) for fruitful discussions about quantification strategies. This work was supported by the German Ministry of Education and Research (BMBF) via the national research node for mass spectrometry in systems medicine MSTARS-2 (16LW0240 and 16LW0239K) to M.S. and N.B. and the German Research Foundation (CRC1588) to N.B. and M.S. Additional funding for M.V.B. was provided by Deutsches Konsortium für Translationale Krebsforschung (DKTK) and the Berlin School for Integrative Oncology (BSIO). B.K was supported by the Deutsche Krebshilfe (DKH, 70114307). Figures 1 and 6 contain elements created in BioRender (Selbach, M. (2025) https://BioRender.com/p98i970)

## Author contributions

M.V.B., B.K., N.B. and M.S. contributed to the design and conceptualisation of this study. The phosphosites in the panel were selected by N.B., B.K. and M.V.B. Experimental work was performed by M.V.B., A.S., N.L., M.H., S.Na. and S.Ni. supervised by P.M., N.B. and M.S. Analysis of experimental data was conducted by M.V.B., B.K. and H.Z. supervised by N.B., M.S. and B.K. The data were mostly visualised by M.V.B. and partly by B.K. The manuscript was written by M.S., M.V.B. and N.B. with editing and contributions from all co-authors.

## Funding

## Competing interests

The authors declare no competing interests.
