## [Transparent Peer Review file · Nature Communications]

Spike-in enhanced phosphoproteomics uncovers synergistic signaling responses to MEK inhibition in colon cancer cells

Corresponding Author: Professor Matthias Selbach

Version 0:

Reviewer comments:

Reviewer #1

(Remarks to the Author)

This very nice study addresses an urgent need for improvement in MS-based proteomics technology. In discovery proteomics, missing values are a main concern, as low abundant proteins and protein modifications often remain undetected. On the other side, it is difficult to distinguish whether undetected proteins or modifications of interest are truly absent (i.e. below limits of detection or quantitation) or simply randomly missed. This manuscript describes a new approach termed SPIED-DIA to boost the detection of low-abundant, phosphorylated peptides of interest using heavy labeled peptide standards in data independent acquisition (DIA) measurements. The method is then applied to investigate mechanisms of treatment resistance in cancer cell lines. This fantastic approach thus enables the authors to investigate low abundant phosphorylation sites, while maintaining the power of explorative proteomics. It adds to a recent trend and growing number of publications using spike-in peptide standards to boost proteomic discovery. These reviewers consider this to be a key step forward for making proteome profiling more robust and better interpretable.

In particular, by combining light and heavy-labeled human cell lysates, the authors demonstrate that the presence of heavy labeled phospho-peptides can enhance the detection of their light counterparts while maintaining good quantification accuracy. This provided the rationale to use heavy spike-in peptides corresponding to 206 clinically relevant phosphorylation sites for SPIED-DIA measurements of human cancer cell lines. About 20% of the target peptides were accurately identified and quantified. Comparisons of heavy to light ratios between positional phosphorylation isomers and raw spectral data indicated correct phospho-site assignment. The new method was then applied to colorectal cancer cell lysates treated with MEK inhibitors to investigate resistance mechanisms to this treatment. In response to MEK inhibition, a previously observed, synergistic increase of AKT phosphorylation was detected in two out of three cell lines. Using SPIED-DIA, this was linked to a concurrent upregulation of JNK Y185 phosphorylation, which was substantiated by the untargeted portion of the SPIED-DIA data. Indeed, a test of MEK and JNK inhibition confirmed the compensatory effect, introducing combined inhibition treatments as a way to prevent therapy resistance.

The authors' main objective, the introduction of heavy-labeled, spike-in peptides as enhancements for DIA detection is original and well substantiated. Still, good quantification accuracy is paramount and should be demonstrated more comprehensively. In particular, we believe that SPIED-DIA's characteristic addition of only few heavy peptides into complex proteomes can be better represented.

- Proper positive and negative controls for target peptide detection are required. The authors show one relevant experiment, where identical amounts of the heavy standard panel were added to decreasing amounts of light cell-lysate. Although they demonstrate decreasing intensities of target signals, the quantification does not fully reflect expected relative intensities (Fig. 1I). We propose to repeat this experiment; some suggestions, making use of species distinction, would be 1) an E. Coli cell lysate is spiked with differing amounts of synthetic, light peptides corresponding to the heavy human phosphopeptide panel. 2) an E.Coli lysate is spiked with differing amounts of a human cell-lysate. For both approaches, the heavy standard panel should be added at identical quantities. A negative control should also be included by omitting the addition of light peptides or human cell-lysate. An explorative SPIED-DIA analysis of the E. Coli component should demonstrate that the heavy spike-in does not perturb consistent identification and quantification of non-target peptides. Similarly, the differing ratios of light and heavy peptides should further confirm quantification accuracy of target peptides.

- One key advantage of targeted proteomic measurements (such as SRM and PRM) is that the "absence" (as defined by limits of detection/quantitation) of a peptide can be clearly demonstrated by the lack of light fragment signals, which contrasts

with the presence of corresponding heavy signals. Since the majority of the 206 target peptides remained undetected, it would be interesting to know whether the absence of a phosphorylation can be visualized via analysis in Skyline, Spectronaut or another software of choice.

Additional remarks from the reviewers are:

- The figures urgently need uniform color coding and organization. For example, color from figure 3A and 3D do not match. In general, the vast amount of different color schemes and legends render many figures rather confusing.
- We appreciate the concise discussion.

Overall, the manuscript presents a novel and intriguing method for combining exploratory and targeted proteomic techniques to study protein phosphorylation. Its application to cancer cell lines reveals a new mechanism and treatment recommendations for overcoming resistance to MEK inhibitors, highlighting the method's significant potential. While the quantification accuracy could be demonstrated more effectively, we believe this manuscript is of high value and look forward to the revised version.

Reviewer #2

(Remarks to the Author)

Reviewer #3

(Remarks to the Author)

Van Bentum et al. describe a method (SPIED-DIA) of using heavy labeled phosphopeptides spiked into a complex phosphopeptide sample to bias for identification of these peptides and their endogenous light unlabeled counterparts in a DIA workflow. They go on to use this method of data acquisition and analysis to propose an interaction between the MEK/ERK pathway and the JNK pathway that could possibly be exploited for multiple drug targeting in certain oncogenic backgrounds.

The method of sample preparation and data acquisition is not novel (save for using an instrument with an ion mobility dimension), and was originally described in Litichevskiy et al., Cell Systems 2018. In that work, the authors used targeted PRM-style fragment-based quantification for the spiked peptides (and their light counterparts) which resulted in a very high re-observation rate for target peptides across a large body of samples (typically over 80% of the spiked peptides and their light counterparts were observed and quantified), but reserved analysis of the rest of the DIA data for a later report. In the present work, the authors rely solely on DIA-based peptide identification with precursor-based quantification. They report a decidedly low observation rate for peptides spiked into the samples. Out of 485 peptides spiked into samples, the maximum number identified in any experiment was only ~70, with the median being closer to ~60 (estimated from Fig. 3). In a quantification workflow that disregards the beacon effects of the heavy peptide spike-ins, the median number of target peptides observed appears to be about ~38. So in summary the authors synthesized 485 peptides and achieved a target observation rate of 10-15%. This is essentially just over a 50% improvement from no spike-ins at all. In my estimation, this seems like a lot of work for marginal returns. I would encourage the authors to explore PRM-style fragment-based quantification to achieve a higher re-observation and quantification rate for their spike-ins, which would make their data sets of higher value to the community.

I also had a little trouble getting behind the claims about MEK and JNK pathway synergies. The authors state: "Importantly, we also observed a significant enrichment of JNK1/2 targets in cluster 6 of HCT116 cells, supporting the results from the targeted analysis (Fig. 5A)." Maybe I'm missing something, but JNK1/2 don't appear very enriched in the figure. Enrichment ratio - which I'm assuming is on a log scale, please clarify - is close to 0 and does not seem significant. Synergy is barely noticeable in the line graph for HCT-116 cluster 6. It's more convincing for DLD-1 Cluster 10, but JNK is not part of that cluster. Overall, the claims about the MEK/JNK synergy seem undersupported by the results presented (despite the fact that they observed a growth rate defect) and the authors should work to strengthen these claims.

In general, the technical aspects of this paper are sound and the experiments are well-executed. However, due to the aforementioned shortcomings and lack of novelty, I can't recommend this manuscript for publication in Nature Communications.

Other suggestions to improve the manuscript are given below:

The authors should make it clearer in the discussion of figure 1 that they are using all the SILAC lights as references for all the corresponding SILAC heavies. At first it appeared to me that you had just spiked in your ~500 spy peptides and got huge benefit, and I was left wondering why. Make it clearer in the methods exactly how the two options were identified and quantified. You refer to "normal SILAC-DIA" in the text, but this phrasing is not found in the methods. Why is the number of precursors in the 1:1 mix so low, though? Need to explain this.

Some of the benefit in identification rates between the large and small libraries seem to arise from FDR differences based on the sizes of the underlying libraries. But were the raw scores different with large vs. small DIA libraries? If not, how do the

authors justify accepting a score as valid in one case and not they other?

I suggest making the Y axis of Figure 11 log₂ rather than log₁₀ for easier assessment.

In the Figure 2C legend, I would suggest to remove the word synergy. Significance stars do not seem to indicate synergy and this is confusing. You're just looking at upregulation, I think? It seems from panel D that there is not synergy between MEKi and EGF stimulation in Caco-2, but it is marked with 3 stars. The related discussion about FGF2 and VEGF-C on page 10 is also confusing.

Growth factor cocktail...interesting. But in real life conditions the growth factors are already present at the time of patient treatment. What justifies the additional stimulation? Wouldn't it be more realistic to grow the cells in rich media and then MEKi? The CaCo-2 data suggest this might be true.

It would help for methodological transparency if an R or Python markdown notebook were available to see how you set up the models and performed data analysis in general. Usually in a linear model you'd denote an interaction term as multiplicative and not just perform a contrast analysis to claim synergy. An appropriately defined linear model would not require contrasts at all. In Supp Fig 5, it is unclear if you set up a multiplicative interaction term in the underlying model before setting up the contrasts. Please clarify exactly how the limma design and contrast matrices were specified.

On page 14, you refer to Supp. Fig. 3 as showing a factor analysis but it does not. Perhaps you meant Supp. Fig. 5?

Figure 5B is a bit confusing. I suggest directing the reader to the interaction column to make your point.

Reviewer #4

(Remarks to the Author)

Summary

In this paper the authors report the development of a mass-spectrometry method, the spike-in enhanced detection in DIA (SPIED-DIA), to detect and quantify a set of commonly assayed phosphorylation sites in signaling pathways. The method can be implemented on standard MS equipment and can be combined with unbiased phosphopeptide detection. The authors take great care in determining detection thresholds, dynamic range and practical utility of the assay testing it in different cell lines. They discover an unexpected synergy between JNK and MEK inhibition in the HCT116 colorectal carcinoma cell line. Thus, the assay seems very useful for signaling studies. As MS based phosphoproteomics usually yields data, where the vast majority of phosphorylation sites is not annotated while many of the known regulatory sites are missing, targeted assays seem the way forward. The reported method is of great interest to the field and can be very helpful in evaluating signaling pathways, however some major and minor concerns should be addressed for the manuscript to be appropriate for publication.

Major points

1. The study is technically well carried out. However, similar approaches have been published before and – laudably - are referenced by the authors (references 24, 25, and 54). It would be helpful for the reader to benchmark SPIED-DIA against one or more of these other methods. Ideally, this should be done experimentally, e.g. using the commercially available peptide standards. If experimental benchmarking is not feasible, the authors should at least discuss the strengths and weaknesses of SPIED-DIA in comparison to the previously published methods.
2. Fig. 1G. There are two rows labelled "SPIED-DIA" in the upset plot.
3. The presentation is a little sloppy with several display items not corresponding to the text (see below).
 - i. Fig. S1C is not included in the text.
 - ii. Fig. S1E. This figure is mentioned in the text, but not included in the supplementary data file.
 - iii. Fig. 1I. This figure is not mentioned in the figure legend.
 - iv. Fig. S3A is supposed to show a factor analysis. There is no figure S3A. There is only a Fig. S3, which shows a validation of SILAC quantification ERK2 Tyr 187 in MS/MS spectra and MS1 traces in chromatography.
 - v. Fig. S3B. There is no figure S3B.
4. Fig. 4A-C. The changes in the activating tyrosine phosphorylation sites of ERK1, ERK2, and JNK1 are very clear. The Table with the heavy spike-in peptides also lists the corresponding activating threonine sites. What is their behavior?
5. To achieve full activation of ERK and JNK both the threonine and the tyrosine in the activation loop need to be phosphorylated, i.e. -pT-X-pY-. The peptide standards do not include such double phosphorylated peptides, but only single phosphorylated versions. That means what is measured are the single phosphorylated forms that are only partially active. This could be confounding the biological interpretation of the results. For instance, the distribution of single vs. double phosphorylated forms can impact cellular transformation of liver cells [Seehofer et al. Context-specific flow through the MEK/ERK module produces cell- and ligand-specific patterns of ERK single and double phosphorylation. *Science Signaling* 9 (413), ra13., 2016]. Thus, it is important to measure both single and double phosphorylated MAPK species.
6. Fig. 6. All inhibitor concentrations should be given.

7. Supplementary Table 1. It would be useful to list the heavy peptides that were consistently detected separately.

8. While it is common that phosphoproteomics studies only measure phosphopeptides and ignore that changes in phosphorylation could reflect changes in protein abundance, it would increase the biological meaningfulness of the results to normalize the abundance of phosphopeptides against the corresponding non-phosphorylated peptides. Thus, a set of peptide standards that utilize both the phosphorylated and non-phosphorylated form of the peptides would be very useful. The non-phosphorylated peptides could for instance be measure in the unbound fraction of the phosphopeptide enrichment step or in a total protein expression profile.

Minor points

1. The kinase activation loop peptide standards (reference 25) are commercially available: <https://www.jpt.com/spiketides-set-kinase-activation-loops-human-heavy/SPT-KAL-L> Line 530: -20C
2. Line 632: 30 minutes instead of 20 (according to results section)
3. For consistency: Caco2 in the whole text and figures (fig. 2 and legend of fig. S9) instead of CaCo2
4. Line 726: only F-test < 0.1 is used not 0.05 (figures 4 and 5).
5. Fig. 3 A & D: suggestion to keep same colors for the different treatments between A and D
6. Fig. 3B: y-axis label unclear
7. Fig. 6 B-D: unit on all x axis - μ M instead of uM
8. Fig. 6 B-D: the 'drug schematics' for MEKi and PI3Ki/JNKi are confusing as their coloring and positioning suggests that they are related to what is shown on the y-axis and the color coded lines. Suggest to remove them.
9. Fig. S3: 'blue the traces' should be 'the blue traces'
10. Fig. S4: 'n-terminal' should be 'N-terminal', and '. modified' should '. Modified'
11. Fig. S5A: 'A' missing in figure
12. Please reorder supplementary figures according to their appearance in text
13. Figs. 5A and 5C are not mentioned in the text.

Reviewer #5

(Remarks to the Author)

Version 1:

Reviewer comments:

Reviewer #1

(Remarks to the Author)

The authors have addressed most of our concerns in a comprehensive and satisfactory way. We appreciate the detailed and well-written discussion of all reviewer comments and the inclusion of our suggested experiments. The newly added negative control nicely demonstrates SPIED-DIAs potential to separate truly from randomly missing peptide detections (Fig. 1I). In the same way, the additional analysis of the titration dataset via Skyline provides interesting insights into chromatographic features of individual targeted peptides (Fig. S5). The differences in quantitation performance between DIA-NN and Skyline are striking and highlight DIA-NN as a software of choice for DIA and, seemingly even more so, for SPIED-DIA experiments (Fig. S3&S4).

A major selling point for SPIED-DIA is its potential to combine targeted quantifications of pre-defined peptides with explorative screens within one measurement. The authors have nicely demonstrated this. One remaining question from our side is the quantitation of phosphopeptides, which were not included into the spike-in panel i.e. which were only analyzed in the explorative part. Is their quantification in DIA-NN in any way influenced by the presence of the heavy spike-in panel?

Minor comments are:

- In Figure 1H, please include a green icon for E.coli to visualize what the green portion in the tubes represents.
- Please show intensities for 0ng in the left part of figure 1I

After these smaller concerns have been addressed, we fully recommend this paper for publication and congratulate the authors to this nice work.

(Remarks on code availability)

Reviewer #2

(Remarks to the Author)

I co-reviewed this manuscript with one of the reviewers who provided the listed reports. This is part of the Nature

Communications initiative to facilitate training in peer review and to provide appropriate recognition for Early Career Researchers who co-review manuscripts.

(Remarks on code availability)

Reviewer #3

(Remarks to the Author)

With respect to your response about the novelty of the method, I agree that it is unclear in Litichevskiy et al. exactly how all data were acquired. However, in another paper that utilized the same data, Vaca Jacome et al. (Nat. Method 17:1237-44, 2020) show how P100 DIA mode data could be interrogated for endogenous analyte counterparts to spiked-in phosphopeptide standards. This demonstrates that the method had been conceived and reduced to practice previously. In that work, the authors were able to achieve "quantification of 92% of all peptides with data completeness of 79%" using their workflow (Fig. 3). Shouldn't that be the benchmark to meet for novelty and/or improvement?

Again, my concerns with this manuscript were not due to the quality of the experiments. They were more about the novelty vis a vis publication in Nature Communications, and whether this particular workflow's efforts justified its return. I applaud the extensive revisions made by the authors to address my other points, and I feel that those revisions have made their conclusions stronger. But not more novel or impactful. Ultimately it is up to the editorial staff to make a judgement about novelty and impact.

(Remarks on code availability)

The code seems OK, but it seems to me that several input files required to replicate the figures in the paper are missing from the code base. So it's not particularly useful from a reproducibility standpoint. I can not comment on the "keyscripts" portion as I do not have test data available to verify functionality of the code.

Reviewer #4

(Remarks to the Author)

The authors have addressed all my comments and also the comments of the other reviewers satisfactorily in their revised paper. I am happy to recommend the paper for publication.

(Remarks on code availability)

Reviewer #5

(Remarks to the Author)

(Remarks on code availability)

Point by point response

We thank all five reviewers for their helpful and constructive feedback! We were pleased to read that they think that our work is a “*very nice study*” that presents a “*novel and intriguing method*” and a “*fantastic approach*” which “*addresses an urgent need*” (reviewers # 1 and 2). We are also happy to hear the reviewers think that the “*technical aspects of this paper are sound*”, that the “*experiments are well-executed*” (reviewer # 3), and that the “*method is of great interest to the field*” and “*very helpful in evaluating signaling pathways*” (reviewers # 4 and 5).

We performed additional experiments and analyses to address the constructive comments by the reviewers. Most notably:

- We performed a new benchmarking experiment with the addition of an *E. coli* spike-in. This experiment confirms the ability of SPIED-DIA to convert absence of evidence for target phosphopeptides (NA) into evidence for low abundance. Using this benchmark, we show that our approach reliably distinguishes between low-abundance target peptides that are present and those that are truly absent. The ability to make more confident absence calls is a key advantage of SPIED-DIA.
- We independently validated SPIED-DIA using a recently commercialized heavy phosphopeptide standard available from Thermo Fisher. This experiment also demonstrates that a commercial and presumably higher-purity standard achieves higher detection rates than our self-made crude standard.
- To study the specific contributions of individual growth factors to synergistic JNK activation in detail, we stimulated HCT116 cells with individual growth factors to discern which factor in the mix contributes to synergistic signalling. This analysis revealed that EGF, FGF and HGF (but not VEGF) stimulate JNK and AKT activation in the presence of MEK inhibition. These results also indicate that the observed signaling response is broad and does not depend on a single growth factor.
- To better characterize kinase activity in the global proteomic data we performed an additional unbiased analysis of kinase activity using the recently published dataset from the Cantley lab based on synthetic peptide libraries (Johnson et al., Nature, 2023). This yielded position-specific scoring matrices (PSSMs) for each kinase, which can be used to assign a likely kinase to about 99% of phosphorylation sites. Using this independent dataset we could validate the signature of synergistic JNK activation in our global phosphoproteomic data.
- We also analysed our data with Skyline. Our results show that our fully automated DIA-NN based workflow is at least on par with semi automatic Skyline-based analysis.

Please find a detailed point by point response below. We think that the additional experiments and analyses further improve the manuscript and we think that it can now be accepted for publication.

Reviewer #1:

This very nice study addresses an urgent need for improvement in MS-based proteomics technology. In discovery proteomics, missing values are a main concern, as low abundant proteins and protein modifications often remain undetected. On the other side, it is difficult to distinguish whether undetected proteins or modifications of interest are truly absent (i.e below limits of detection or quantitation) or simply randomly missed. This manuscript describes a new approach termed SPIED-DIA to boost the detection of low-abundant, phosphorylated peptides of interest using heavy labeled peptide standards in data independent acquisition (DIA) measurements. The method is then applied to investigate mechanisms of treatment resistance in cancer cell lines. This fantastic approach thus enables the authors to investigate low abundant phosphorylation sites, while maintaining the power of explorative proteomics. It adds to a recent trend and growing number of publications using spike-in peptide standards to boost proteomic discovery. These reviewers consider this to be a key step forward for making proteome profiling more robust and better interpretable.

In particular, by combining light and heavy-labeled human cell lysates, the authors demonstrate that the presence of heavy labeled phospho-peptides can enhance the detection of their light counterparts while maintaining good quantification accuracy. This provided the rationale to use heavy spike-in peptides corresponding to 206 clinically relevant phosphorylation sites for SPIED-DIA measurements of human cancer cell lines. About 20% of the target peptides were accurately identified and quantified. Comparisons of heavy to light ratios between positional phosphorylation isomers and raw spectral data indicated correct phospho-site assignment. The new method was then applied to colorectal cancer cell lysates treated with MEK inhibitors to investigate resistance mechanisms to this treatment. In response to MEK inhibition, a previously observed, synergistic increase of AKT phosphorylation was detected in two out of three cell lines. Using SPIED-DIA, this was linked to a concurrent upregulation of JNK Y185 phosphorylation, which was substantiated by the untargeted portion of the SPIED-DIA data. Indeed, a test of MEK and JNK inhibition confirmed the compensatory effect, introducing combined inhibition treatments as a way to prevent therapy resistance.

The authors' main objective, the introduction of heavy-labeled, spike-in peptides as enhancements for DIA detection is original and well substantiated. Still, good quantification accuracy is paramount and should be demonstrated more comprehensively. In particular, we believe that SPIED-DIAs characteristic addition of only few heavy peptides into complex proteomes can be better represented.

We thank the reviewer for the encouraging feedback! We performed additional experiments and analyses to better represent SPIED-DIAs characteristic addition of only a few heavy peptides into complex proteomes.

1. Proper positive and negative controls for target peptide detection are required. The authors show one relevant experiment, where identical amounts of the heavy standard panel were

added to decreasing amounts of light cell-lysate. Although they demonstrate decreasing intensities of target signals, the quantification does not fully reflect expected relative intensities (Fig. 1I). We propose to repeat this experiment; some suggestions, making use of species distinction, would be 1) an *E. Coli* cell lysate is spiked with differing amounts of synthetic, light peptides corresponding to the heavy human phosphopeptide panel. 2) an *E. Coli* lysate is spiked with differing amounts of a human cell-lysate. For both approaches, the heavy standard panel should be added at identical quantities. A negative control should also be included by omitting the addition of light peptides or human cell-lysate. An explorative SPIED-DIA analysis of the *E. Coli* component should demonstrate that the heavy spike-in does not perturb consistent identification and quantification of non-target peptides. Similarly, the differing ratios of light and heavy peptides should further confirm quantification accuracy of target peptides.

This is a valuable suggestion, and we performed additional experiments and analyses to address it (see below). First, however, we would like to point out that there is a 2nd relevant experiment already present, shown in Fig. 1 G. In this experiment, we added our heavy synthetic phosphopeptide panel to HEK293 cell lysate, performed phosphopeptide enrichment and measured samples using DIA-PASEF. We then compared the number of identified phosphopeptides in this experiment to a standard label-free analysis. These data show that adding the heavy spike-in standard improves detection of target phosphorylation sites.

The suggestion to repeat the experiment shown in Fig. 1H and I with an additional *E. coli* spike-in is excellent, and we did exactly what the reviewer suggested. The addition of *E. coli* allowed us to perform the requested negative control, that is to omit the human lysate. Importantly, SPIED-DIA does not detect any light phosphopeptides in this sample based on our filter criteria (new Fig. 1 I, right panel). Furthermore, the data show that SPIED-DIA adequately quantifies the changing amounts of the light human phosphopeptides across the dilution series. As expected and consistent with figure 1 E, the accuracy and precision of quantification is a function of peptide intensity, with more intense peptides generally providing better quantification. We added the new figure and the following text to the revised manuscript:

“As negative control we also added the same amount of E. coli to each sample (Fig. 1H).”

“In the negative control, no peptides passed our filtering criteria, highlighting the specificity of our workflow (Fig. 1 I). Consequently, our approach reliably distinguishes between low-abundance target peptides that are present and those that are truly absent.”

2. One key advantage of targeted proteomic measurements (such as SRM and PRM) is that the “absence” (as defined by limits of detection/quantitation) of a peptide can be clearly demonstrated by the lack of light fragment signals, which contrasts with the presence of corresponding heavy signals. Since the majority of the 206 target peptides remained undetected, it would be interesting to know whether the absence of a phosphorylation can be visualized via analysis in Skyline, Spectronaut or another software of choice.

We fully agree: A key advantage of the H spike-in is indeed that we can more confidently draw conclusions about the absence of corresponding light endogenous peptides. Specifically, the inclusion of a heavy spike-in uncouples detection from quantification. Therefore, when the heavy spike-in is consistently detected, it is unlikely that the corresponding light target peptide is missed for purely technical reasons.

The new experiment with an *E. coli* spike-in and the omission of human lysate (updated Figure 1 H and I) allows us to specifically address this question more thoroughly. As suggested by the reviewer, we analysed these data using both our automated DIA-NN based workflow (Fig. S3) and Skyline (Fig. S4). As expected, the quality of quantification depends on the abundance of the target peptide: For peptides with higher abundance in the light channel we see clear separation across the dilution series. Importantly, for these peptides we also see that their log₂(L/H) ratio is consistently lower in the negative control than in the samples containing minute amounts of HEK lysate. This highlights the advantage of SPIED-DIA to transform an “absence of evidence” (NA) into an “evidence of low abundance”.

To explain this in more detail, we added the following paragraph to the Results section:

“Missing values pose a significant challenge in quantifying changes in peptide abundance across conditions, as they can either represent truly absent peptides or peptides that were missed for technical reasons. In both scenarios, the result is an NA value that cannot be used for reliable comparisons across conditions. The inclusion of a heavy spike-in uncouples detection from quantification, providing a key advantage: when the heavy spike-in is consistently detected, it is unlikely that the corresponding light target peptide is missed for purely technical reasons. Instead, the absence of the light peptide can be more reliably attributed to its genuine absence or low abundance. Making more confident absence calls is a key advantage of targeted proteomic methods like SPIED-DIA. We leveraged this feature to improve quantification of target peptides across conditions by rescuing data in cases where the light peptide was not detected but the heavy peptide was. To implement this advantage, we implemented specific filtering criteria: For a peptide to be used for quantification, it needs to be consistently detected as a heavy spike-in across all samples. Additionally, the light target peptide is required to pass filtering criteria in at least two out of three replicates in at least one experimental condition. This ensures that comparisons across samples reflect changes between signal and signal (light target peptide detected in both conditions) or signal and background noise (light target peptide detected in only one condition), but not background noise and background noise (light target peptide not detected in either condition).”

Additional remarks from the reviewers are:

3. The figures urgently need uniform color coding and organization. For example, color from figure 3A and 3D do not match. In general, the vast amount of different color schemes and legends render many figures rather confusing.

We fully agree and apologize for this. We now make sure that the coloring schemes are consistent across figures.

4. We appreciate the concise discussion.

Thanks!

5. Overall, the manuscript presents a novel and intriguing method for combining exploratory and targeted proteomic techniques to study protein phosphorylation. Its application to cancer cell lines reveals a new mechanism and treatment recommendations for overcoming resistance to MEK inhibitors, highlighting the method's significant potential. While the quantification accuracy could be demonstrated more effectively, we believe this manuscript is of high value and look forward to the revised version.

Thanks a lot for the encouraging comments! We hope you that the additional experiments and analyses provide further support that the SPIED-DIA can yield good quantitative accuracy for target phosphopeptides that would otherwise escape detection.

Reviewer #2 (Remarks to the Author):

Thanks for your help!

Reviewer #3 (Remarks to the Author):

Van Bentum et al. describe a method (SPIED-DIA) of using heavy labeled phosphopeptides spiked into a complex phosphopeptide sample to bias for identification of these peptides and their endogenous light unlabeled counterparts in a DIA workflow. They go on to use this method of data acquisition and analysis to propose an interaction between the MEK/ERK pathway and the JNK pathway that could possibly be exploited for multiple drug targeting in certain oncogenetic backgrounds.

The method of sample preparation and data acquisition is not novel (save for using an instrument with an ion mobility dimension), and was originally described in Litichevskiy et al., Cell Systems 2018. In that work, the authors used targeted PRM-style fragment-based quantification for the spiked peptides (and their light counterparts) which resulted in a very high re-observation rate for target peptides across a large body of samples (typically over 80% of the spiked peptides and their light counterparts were observed and quantified), but reserved analysis of the rest of the DIA data for a later report.

We thank the reviewers for this feedback. We fully agree that our method is related to existing methods, and we cited all of the relevant ones we found, as pointed out by reviewer # 4 ("laudably - are referenced by the authors (references 24, 25, and 54)"). We now added a reference to Litichevskiy et al. as well. In fact, the "P100" method employed in this paper is itself not novel but actually based on a publication by Aebelin and co-workers (MCP, 2016). We also added a reference to this paper.

Despite the overall similar idea to use targeted phosphoproteomics of particularly relevant phosphopeptides to assess cellular signalling states we would like to point out that our SPIED-DIA method differs in more aspects than the use of an instrument with ion mobility dimension. Based on the information provided in the paper, it appears that Litichevskiy et al. did not perform a targeted analysis of the DIA data in the way we do it in SPIED-DIA. The material and methods section states:

*“Peptides were separated on a C18 column (EASY-nLC 1000, Thermo Scientific) and subsequently **analyzed by mass spectrometry (MS) as described in Aebelin et al., or in DIA mode** (Q ExactiveTM-HF OrbitrapTM, Thermo Scientific).”*

The corresponding methods section in Aebelin et al. paper states:

*“Each full scan was followed by **fully scheduled, targeted HCD MS/MS scans** at resolution 17,500 (Isolation width 2 m/z, ACG Target 2e5, 50 ms Max IT). Each peptide species was subjected to targeting MS/MS for 3–5 min depending on the empirical chromatographic properties, centered on the average observed retention time of two scheduling runs containing synthetic versions of a subset of isotopically labeled phosphopeptide probes.”*

It thus appears that Litichevskiy et al. measure every sample twice, once in PRM and once in DIA mode. This is an interesting alternative to our “DIA only” approach that provides certain advantages (such as the better detection rate of target phosphosite) but also has disadvantages (longer acquisition time, complexity of setting up scheduled PRM assays etc.). PRM requires method development time and separate MS runs. Given the current trend in the field towards DIA, the specific methodological aim of our study is to improve detection in DIA via a simple and straightforward way. Other methods (PRM, SureQuant, Pseudo-PRM, TOMAHAQ, Scout-MRM) may yield improved coverage of target peptides but are more difficult to set up than DIA.

Other differences between our paper and the paper by Litichevskiy and our work are that we used a completely automated data processing workflow based on DIA-NN while Aebelin and Litichevskiy et al. used a semi-automated workflow using Skyline. To highlight this point, we performed a direct comparison of our DIA-NN-based workflow and a Skyline analysis (new Fig. S4). This analysis reveals the SPIED-DIA workflow is not only simpler but provides overall better quantification than the Skyline-based approach. In addition to these methodological differences the biological focus and findings were also different.

In the present work, the authors rely solely on DIA-based peptide identification with precursor-based quantification. They report a decidedly low observation rate for peptides spiked into the samples. Out of 485 peptides spiked into samples, the maximum number identified in any experiment was only ~70, with the median being closer to ~60 (estimated from Fig. 3). In a quantification workflow that disregards the beacon effects of the heavy peptide spike-ins, the median number of target peptides observed appears to be about ~38. So in summary the authors synthesized 485 peptides and achieved a target observation rate of 10-15%. This is

essentially just over a 50% improvement from no spike-ins at all. In my estimation, this seems like a lot of work for marginal returns. I would encourage the authors to explore PRM-style fragment-based quantification to achieve a higher re-observation and quantification rate for their spike-ins, which would make their data sets of higher value to the community.

We fully agree with the reviewer that more complex targeted acquisition methods like PRM, SureQuant, Pseudo-PRM, TOMAHAQ, Scout-MRM yield a higher observation rate for target peptides. We clearly point this out in the Discussion section:

“Despite these advantages, SPIED-DIA also has limitations. SPIED-DIA provides only a modest sensitivity boost for target peptides because it lacks the longer selective ion collection periods found in other targeted approaches”

In addition, we would also like to clarify that we spiked in 485 phosphopeptides. The phosphopeptides were synthesized by JPT in a pooled set-up (SpikeMix™ Peptide Pools, <https://www.jpt.com/products-services/peptide-pools/>). While this is cost-effective, there is no guarantee that each peptide is adequately represented in the pool. Therefore, while we submitted a list of 485 unique phosphopeptides to be synthesized (as described in the Material and Methods section), this does not mean that we actually spiked-in 485 phosphopeptides. We already mentioned this issue as a limitation of our study in the Discussion section:

“To economize, we opted for pooled synthesis of heavy reference peptides. However, this approach led to unsuccessful synthesis of some desired peptides, rendering them unusable in our targeted strategy.”

We also specifically mentioned this point in the Results section:

“Although we included several AKT1 phosphorylation sites among our target peptides, the synthesis of heavy peptides failed, which explains their absence in the targeted data (Supplemental Table 1).”

To make this point clearer in the Material and Methods section we now added the following sentence to the Material and Methods section:

“These SpikeMix™ Peptide Pools are more cost-effective than synthesizing individual peptides; however, it is important to note that all peptides on the list are anticipated to be synthesized successfully with adequate yields.”

In addition, we added this text to the Results section:

“Selected sites were mapped to an in silico tryptic digest of the human proteome and 485 peptides meeting criteria for synthesis were ordered as SpikeMix™ Peptide Pools. Analysing this heavy synthetic peptide pool via single shot DDA analysis (see Materials and Methods for

details) we identified 240 peptides that we used to create a heavy phosphopeptide library (Fig. S1E, Supplementary table 1).

To clarify the process of peptide synthesis, detection of the heavy peptides in DDA and DIA and detection of endogenous light peptides we added the following figure as a new supplementary figure S1E:

Finally, we would like to point out that the detection of endogenous phosphopeptides depends on the signalling state of the cells. For example, in the benchmarking experiment described in Fig. 1 G we used non-stimulated HEK293 cells. In these experiments we detected 173 heavy spike-in peptides, including many sites not expected to be significantly phosphorylated in HEK293 cells under baseline conditions. This may explain why we indeed only detected 40 endogenous target peptides. It is perhaps not surprising that analysing a diverse range of cancer cell models (breast, lung, pancreatic, prostate, skin) as done by Litichevskiy and co-workers yields a higher fraction of target IDs. Importantly, however, the 40 endogenous targets we detected in HEK cells using SPIED-DIA is still substantially higher than the 24 peptides we detected with the label free workflow in the same cell model.

To independently evaluate the improved coverage of target peptides, we utilized a recently commercialized heavy phosphopeptide standard available from Thermo Fisher. While this standard contains 135 phosphopeptides in total, we focused on the 81 peptides that contain a single C-terminal lysine or arginine residue, as these are compatible with our DIA-NN workflow based on SILAC labels. Among these, we detected 72 peptides via DDA to construct a spectral library. Following one of the benchmarking experiments described in our paper, we spiked the heavy phosphopeptides into varying amounts of HEK-derived phosphopeptides. SPIED-DIA successfully identified 24 endogenous (light) target peptides with good quantification across the dilution series. This experiment independently validates SPIED-DIA using a commercial phosphopeptide standard. Furthermore, the data indicate that the commercially available, and presumably higher-purity, heavy phosphopeptide standard achieves superior detection rates for

both heavy (60/81 = 74%) and light target peptides (24/81 = 30%) compared to our self-made crude heavy standard (36% and 8%, respectively). Since the main focus of our manuscript is on the self-made standard and the paper is already extensive, we decided not to include these additional results. However, we do discuss the potential applications of this and other commercially available standards in the manuscript's discussion section.

I also had a little trouble getting behind the claims about MEK and JNK pathway synergies. The authors state: “Importantly, we also observed a significant enrichment of JNK1/2 targets in cluster 6 of HCT116 cells, supporting the results from the targeted analysis (Fig. 5A).” Maybe I’m missing something, but JNK1/2 don’t appear very enriched in the figure. Enrichment ratio - which I’m assuming is on a log scale, please clarify - is close to 0 and does not seem significant. Synergy is barely noticeable in the line graph for HCT-116 cluster 6. It’s more convincing for DLD-1 Cluster 10, but JNK is not part of that cluster. Overall, the claims about the MEK/JNK synergy seem undersupported by the results presented (despite the fact that they observed a growth rate defect) and the authors should work to strengthen these claims.

This point is well taken. We fully agree that the evidence for enrichment of JNK targets in the global analysis is not very strong. This actually highlights a key strength of the targeted analysis: Identifying a synergistic increase in JNK1_Y185 phosphorylation allowed us to directly infer that JNK activity is changing since this site is indicative of JNK activity. Observing a change in the abundance of this single but highly informative phosphopeptide provides direct evidence for a change in JNK activity. The global data “only” serves as an additional means to validate this finding independently, using *independent* phosphorylation sites.

We thank the reviewer for carefully checking the figure! Indeed, there was an issue with the scale which we now corrected. The actual enrichment ratios for JNK1 and JNK2 are 2.5 and 2.9, and we now present them in log10 scale. We also provide details on how exactly these ratios are calculated in the github associated with the manuscript (the relevant script can be found under `Scripts_Manuscript/CellLinePanel_Global/Cluster_Kinase_Enrichment.R`).

The modest statistical significance for the enrichment of annotated JNK substrates in cluster 6 (enrichment p-values for JNK1 and 2 are 0.13 and 0.097, respectively) is to some extent due to the poor annotation of kinase substrates. This is especially the case for PhosphoSitePlus, which shows a strong bias toward few well-characterized kinases (e.g., CDK1, PRKCA, CK2A1), while most other kinases have only a small number of assigned sites. To improve coverage, we used our *In Vitro Kinase-to-Phosphosite Database* (iKiP-DB) (Mari et al., JPR, 2022). However, iKiP-DB is exclusively based on an *in vitro* kinase dataset and therefore also does not comprehensively capture true substrates of a given kinase. Indeed, manual inspection of the phosphorylation sites in cluster 6 identifies additional JNK substrates such as SPAG9_T217 (FC: 1.13, psyn-val: 0.1) and ATF_S112 (FC: 1.13, psyn-val: 0.1).

Having said this, we agree with the reviewer that it is important to provide additional support for synergistic activation of JNK in the global phosphoproteomic data. To this end, we used a recently published dataset from the Cantley lab (Johnson et al., Nature, 2023). In this study, the authors used synthetic peptide libraries to profile the substrate sequence specificity of 303 purified recombinant Ser/Thr kinases. This yielded position-specific scoring matrices (PSSMs) for each kinase, which can be used to assign a likely kinase to about 99% of phosphorylation sites. We used these data to screen for evidence of changes in kinase activity in the global phosphoproteomic data. Reassuringly, this analysis revealed synergistic activation of JNK in HCT116 cells (new figure 5C).

C

We added this figure and the following paragraph to the manuscript:

“A recent study employed synthetic peptide libraries and in vitro kinase assays to systematically profile the substrate specificities of the human serine/threonine kinome, generating a comprehensive atlas that enables the prediction of kinase-substrate relationships (Johnson et al., Nature, 2023). Leveraging this independent dataset, we analyzed the enrichment of kinase motifs in the factor analysis results of our global phosphoproteomic data. This analysis revealed significant enrichment of JNK motifs in HCT116 cells, confirming a signature of synergistic JNK activation in the global dataset (Fig. 5C).”

In general, the technical aspects of this paper are sound and the experiments are well-executed. However, due to the aforementioned shortcomings and lack of novelty, I can't recommend this manuscript for publication in Nature Communications.

We think that the additional data and analyses further improved the quality of our manuscript. Also, while we agree that there are other useful targeted phosphoproteomic studies, we think that our SPIED-DIA approach provides key advantages. The ability to use the commercially available multipathway phosphopeptide standard will further broaden the applicability of our method. Finally, the finding that an independent kinase-substrate dataset confirms synergistic JNK activation further strengthens our biological conclusions.

Other suggestions to improve the manuscript are given below:

The authors should make it clearer in the discussion of figure 1 that they are using all the SILAC lights as references for all the corresponding SILAC heavies. At first it appeared to me that you had just spiked in your ~500 spy peptides and got huge benefit, and I was left wondering why. Make it clearer in the methods exactly how the two options were identified and quantified. You refer to “normal SILAC-DIA” in the text, but this phrasing is not found in the methods. Why is the number of precursors in the 1:1 mix so low, though? Need to explain this.

We thank the reviewer for this comment. We added the following text to the Results section:

“Importantly, when applying the final workflow, a single DIA raw data file is used to perform two distinct types of analyses: one that generates global untargeted label-free quantified data and another, the SPIED-DIA analysis, which allows for more sensitive detection and stable isotope-based quantification of targeted peptides.”

Additionally, we added the sentence:

“We note that in this experiment, we are using all the SILAC lights as references for all the corresponding SILAC heavies.”

Some of the benefit in identification rates between the large and small libraries seem to arise from FDR differences based on the sizes of the underlying libraries. But were the raw scores different with large vs. small DIA libraries? If not, how do the authors justify accepting a score as valid in one case and not the other?

We appreciate this insightful question and have performed additional analyses to rigorously address the concern regarding potential FDR differences caused by library size. Specifically, we aimed to determine whether the improvements observed under the SPIED-DIA workflow arise from genuine enhancements in detection confidence provided by the spike-in peptides or merely from the reduced library size.

We start by recapitulating the data presented in Figure 1G. Of the 240 peptides in the SPIED-DIA target library, 85 were also present in the LFQ-HpH library. In the LFQ-HpH workflow, only 24 target peptides were identified. The SPIED-DIA workflow identified 40 target peptides, 29 of which were present in the LFQ-HpH library. Notably, 5 peptides present in the LFQ-HpH library were only identified when using the SPIED-DIA approach, highlighting improved detection enabled by the spike-in peptides.

To directly test whether library size alone accounts for the observed differences, we constructed a reduced LFQ-HpH library (~240 peptides), composed of 85 peptides overlapping with the synthetic set and an additional 155 peptides confidently identified in LFQ-HpH runs. We observed that identifications were markedly lower: 16 peptides with Match-Between-Runs (MBR) and 19 peptides without MBR, at the same confidence thresholds. Next, we directly compared 76 overlapping phosphopeptide sequences (79 precursors) between SPIED-DIA and the reduced LFQ-HpH-like library workflow across key scoring metrics: Q.Value (global FDR), PTM.Q.Value (post-translational modification FDR), and PTM.Site.Confidence (site localization confidence). Across all these metrics, the SPIED-DIA workflow consistently showed lower (that is, better) Q.Values, indicating improved confidence in peptide identifications (see fig. A and B below). The differences in Q.Values correlated with higher MS1 areas for peptides in the SPIED-DIA workflow (fig. C below). This suggests that the spiked-in peptides serve as “beacons”, enhancing detection confidence and enabling more reliable identification of target peptides. Importantly, this improvement cannot be attributed solely to FDR effects resulting from library size, as libraries of similar size (SPIED-DIA vs. reduced LFQ-HpH) demonstrated clear performance differences in peptide identification and scoring.

To further validate the role of spike-ins, we also analyzed samples using the small spike-in library alone but without adding the heavy spike-in peptides. This approach failed to identify any precursors within the highly complex background, reinforcing that the presence of the heavy spike-in is necessary to achieve the observed gains in detection confidence under the SPIED-DIA workflow.

We conclude that the improvements observed in the SPIED-DIA workflow are not solely driven by FDR differences due to library size. Instead, the spike-in peptides act as high-abundance beacons that enhance detection confidence, as demonstrated by consistently better Q.Values, higher MS1 areas, and the inability of a size-matched LFQ-HpH library to replicate the performance. We thank the reviewer for raising this point, as it enabled us to perform additional analyses that further clarified the unique advantages of the SPIED-DIA workflow. However, we did not include these additional results in the manuscript to avoid further increasing the complexity of an already detailed analysis, which could compromise the overall readability.

I suggest making the Y axis of Figure 11 log₂ rather than log₁₀ for easier assessment.

Agreed. We changed the y-axis as suggested.

In the Figure 2C legend, I would suggest to remove the word synergy. Significance stars do not seem to indicate synergy and this is confusing. You're just looking at upregulation, I think? It seems from panel D that there is not synergy between MEKi and EGF stimulation in Caco-2, but it is marked with 3 stars. The related discussion about FGF2 and VEGF-C on page 10 is also confusing.

We thank the reviewer for this sharp observation and for pointing out the inconsistency in Figure 2C. Upon careful re-evaluation, we identified an error that occurred during the post-processing of the figure, which led to the significance stars not accurately reflecting the receptor-driven

synergistic upregulation as defined in the methods section (paragraph “Cell line panel screen for MEKi-dependent receptor-mediated feedbacks”).

We have now corrected Figure 2C to display the appropriate significance grading and ensured alignment with the synergy criteria described in the methods. We are grateful for the opportunity to clarify this point and believe that this correction resolves the confusion regarding the discussion of FGF2 and VEGF-C on page 10, as well as the apparent discrepancy in the Caco2 results relative to Figure 2D.

Thank you again for your attention to detail, which allowed us to improve the accuracy and clarity of our work.

Growth factor cocktail...interesting. But in real life conditions the growth factors are already present at the time of patient treatment. What justifies the additional stimulation? Wouldn't it be more realistic to grow the cells in rich media and then MEKi? The CaCo-2 data suggest this might be true.

We appreciate the reviewer's point that acute stimulation after starvation does not fully replicate *in vivo* conditions. However, it is worth noting that the continuous and homogeneous supply of rich media in tissue culture also fails to capture the complexity of intercellular communication in human tumors. Like all *in vitro* models, tissue culture systems present trade-offs: their simplicity, ease of interpretation, and manipulability make them powerful tools, but they inherently lack some of the physiologically relevant aspects of cancer biology. In practice, acute stimulation of tissue culture cells with growth factors has been a cornerstone approach in cell signaling research, leading to the discovery of many key pathways. This experimental setup is particularly advantageous for disentangling receptor-mediated feedback mechanisms from receptor-independent ones. Furthermore, the signal-to-noise ratio is typically higher in acute stimulation experiments following starvation, making them highly informative and widely used in the field. Finally, we would like to point out that the synergistic signalling response we identified in our acute stimulation experiments were validated in a continuous growth experiment shown w/o starvation in Figure 6. Thus, our findings appear to also be relevant for continuous growth conditions. Finally, as we mentioned in the discussion, a recent study showed that the KRAS inhibitor sotorasib and the MAP2K4 inhibitor HRX-0233 synergistically inhibit growth of KRAS mutant tumors in mouse xenografts (Jansen et al., PNAS, 2024). While mouse xenografts still do not recapitulate the complexity of human tumors, these data still indicate that synergistically targeting MEK/ERK and JNK signaling could be a viable approach in cancer therapy.

We also acknowledge that using a cocktail of growth factors may be considered unconventional. However, this approach offers a significant advantage by probing the involvement of multiple receptors simultaneously, thereby expanding the search space and increasing the likelihood of identifying relevant feedback mechanisms. To further dissect the contribution of individual growth factors and their receptors to the synergistic signaling responses observed in HCT116 cells, we conducted additional experiments. Specifically, we treated cells with the MEK inhibitor selumetinib and stimulated them individually with EGF, HGF, FGF2, or VEGF-C, or no stimulus

(BSA control). These experiments demonstrated that EGF, HGF, and FGF2—but not VEGF-C—signaling can mediate synergistic JNK activation. This finding indicates that the observed signaling response is broad and does not rely on a single growth factor.

We added the following text and the supplemental figure to the Results section:

“The experiments described thus far were conducted using a cocktail of growth factors. To investigate the specific contributions of individual growth factors to synergistic JNK activation in HCT116 cells, we performed additional stimulation experiments using single growth factors. For this purpose, cells were pretreated with a MEK inhibitor and then stimulated individually with EGF, HGF, FGF2, or VEGF-C, or solvent control BSA(Fig. S15A). Both targeted (Fig. S15B) and global SPIED-DIA analysis (Fig. S15C, D) revealed that EGF, HGF, and FGF2— but not VEGF-C—are capable of mediating synergistic JNK activation. This finding suggests that the observed signaling response is broad and does not depend on a single growth factor.”

It would help for methodological transparency if an R or Python markdown notebook were available to see how you set up the models and performed data analysis in general. Usually in a linear model you'd denote an interaction term as multiplicative and not just perform a contrast analysis to claim synergy. An appropriately defined linear model would not require contrasts at all. In Supp Fig 5, it is unclear if you set up a multiplicative interaction term in the underlying model before setting up the contrasts. Please clarify exactly how the limma design and contrast matrices were specified.

The reviewer is right that if we would use standard R linear models, the significance reported by 'lm()' for the interaction term would be the correct way to quantify this. In this study we use limma, which has the main advantage that it uses empirical bayes estimates of variance, which is superior in performance when limited replicates are used in omics analysis, and therefore standard in the field. Within the context of limma, the statistical test that is appropriately modelling the interaction is based on contrasts between differences. For the statistical details, we would refer to a paper that describes the appropriate use of contrasts for interaction models [doi:10.12688/f1000research.27893.1](https://doi.org/10.12688/f1000research.27893.1):

For transparency, the entire code is now also available on Github (<https://github.com/Mirjamva/SPIED-DIA>)

For ease of reviewing, we show the relevant code snippet here (HCT116):

Limma code:

```
# define experimental design
ligandmix_levels <- c(rep("BSA", 6), rep("LigandMix", 6))
meki_levels <- c(rep("DMSO", 3), rep("MEKi", 3),
                rep("DMSO", 3), rep("MEKi", 3))
```

```
Ligandmix <- factor(ligandmix_levels, levels = c("BSA", "LigandMix"))
```

```

MEKi <- factor(meki_levels, levels = c("DMSO", "MEKi"))

TS <- factor(paste(Ligandmix, MEKi, sep = "."),
            levels = unique(paste(ligandmix_levels, meki_levels, sep = ".")))

design <- model.matrix(~0 + TS)
colnames(design) <- levels(TS)

# Define Contrasts
contrast_matrix_hct116 <- makeContrasts(
  LMvsBSAinDMSO = LigandMix.DMSO - BSA.DMSO,
  LMvsBSAinMEKi = LigandMix.MEKi - BSA.MEKi,
  SynSign = (LigandMix.MEKi - BSA.MEKi) - (LigandMix.DMSO - BSA.DMSO),
  MEKivsDMSOinBSA = BSA.MEKi - BSA.DMSO,
  MEKivsDMSOinLM = LigandMix.MEKi - LigandMix.DMSO,
  SynSign2 = (LigandMix.MEKi - LigandMix.DMSO) - (BSA.MEKi - BSA.DMSO),
  levels = design
)

# Fit the Model and Apply Contrasts
fit_hct116 <- lmFit(hct116_data, design)
fit_hct116 <- contrasts.fit(fit_hct116, contrast_matrix_hct116)
fit2_hct116 <- eBayes(fit_hct116)

```

On page 14, you refer to Supp. Fig. 3 as showing a factor analysis but it does not. Perhaps you meant Supp. Fig. 5?

Thanks, checked and changed

Figure 5B is a bit confusing. I suggest directing the reader to the interaction column to make your point.

Point well taken, highlighted interaction term in the fig

Reviewer #4 (Remarks to the Author):

Summary

In this paper the authors report the development of a mass-spectrometry method, the spike-in enhanced detection in DIA (SPIED-DIA), to detect and quantify a set of commonly assayed phosphorylation sites in signaling pathways. The method can be implemented on standard MS equipment and can be combined with unbiased phosphopeptide detection. The authors take

great care in determining detection thresholds, dynamic range and practical utility of the assay testing it in different cell lines. They discover an unexpected synergy between JNK and MEK inhibition in the HCT116 colorectal carcinoma cell line. Thus, the assay seems very useful for signaling studies. As MS based phosphoproteomics usually yields data, where the vast majority of phosphorylation sites is not annotated while many of the known regulatory sites are missing, targeted assays seem the way forward. The reported method is of great interest to the field and can be very helpful in evaluating signaling pathways, however some major and minor concerns should be addressed for the manuscript to be appropriate for publication.

Major points

1. The study is technically well carried out. However, similar approaches have been published before and – laudably - are referenced by the authors (references 24, 25, and 54). It would be helpful for the reader to benchmark SPIED-DIA against one or more of these other methods. Ideally, this should be done experimentally, e.g. using the commercially available peptide standards. If experimental benchmarking is not feasible, the authors should at least discuss the strengths and weaknesses of SPIED-DIA in comparison to the previously published methods.

We thank the reviewer for the supportive comments and agree that benchmarking SPIED-DIA against other methods would be an interesting avenue for exploration. However, benchmarking against methods such as PRM or SureQuant is very laborious as it requires adjusting acquisition parameters for each target peptide. The ease of setting up the method is actually a key advantage of SPIED-DIA. Also, we do already discuss strengths and weaknesses of different targeted acquisition methods in the discussion section. We recently reviewed advanced targeted acquisition strategies (<https://doi.org/10.1016/j.mcpro.2021.100165>) and feel that discussing this in detail is beyond the scope of this primary research paper.

However, we did follow the suggestion of the reviewer and performed additional experiments with the commercially available multipathway heavy phosphopeptide standard. This standard contains 135 phosphopeptides, of which 81 contain a single C-terminal lysine or arginine residue and are therefore compatible with our DIA-NN workflow based on SILAC labels. 72 of these peptides were detected via DDA and used to construct a spectral library. Following the experiment shown in figure 1 h, we benchmarked the performance of this spike-in in a dilution series experiment (show fig. R1 below). In these experiments, we detected 60 of the (heavy) spike-in peptides and 24 of the corresponding (light) target peptides with overall good quantitative performance across the dilution series. Although this experiment does not benchmark different targeted acquisition methods, it validates SPIED-DIA with a commercial standard. This experiment also shows that a commercial and presumably higher-purity heavy phosphopeptide standard achieves superior detection rates for both heavy ($60/81 = 74\%$) and light target peptides ($24/81 = 30\%$) compared to our self-made crude SpikeMix™ Peptide Pool standard. Since the rest of the manuscript focuses on our self-made standard and is already quite long we decided not to include these data in the paper.

2. Fig. 1G. There are two rows labelled “SPIED-DIA” in the upset plot.

This is correct. The upper row displays IDs obtained from the light sample with the heavy spike-in, while the lower row shows IDs from the control sample containing only the heavy spike-in. This approach estimates false positives, with the relevant information presented in the blue, red, or grey boxes.

3. The presentation is a little sloppy with several display items not corresponding to the text (see below).

We sincerely thank the reviewer for carefully checking our manuscript and bringing these discrepancies to our attention. We apologize for these oversights and greatly appreciate the opportunity to correct them.

i. Fig. S1C is not included in the text. Mentioned in the text

ii. Fig. S1E. This figure is mentioned in the text, but not included in the supplementary data file. Now included

iii. Fig. 1I. This figure is not mentioned in the figure legend. Now mentioned

iv. Fig. S3A is supposed to show a factor analysis. There is no figure S3A. There is only a Fig. S3, which shows a validation of SILAC quantification ERK2 Tyr 187 in MS/MS spectra and MS1 traces in chromatography. This is now figure S7A and correctly linked in the text

v. Fig. S3B. There is no figure S3B. Cross checked all figure references in the text.

4. Fig. 4A-C. The changes in the activating tyrosine phosphorylation sites of ERK1, ERK2, and JNK1 are very clear. The Table with the heavy spike-in peptides also lists the corresponding activating threonine sites. What is their behavior?

This is another valid point. These threonine sites were indeed included in the list of peptides to be synthesized in the SpikeMix™ Peptide Pool. Specifically, we ordered these peptides:

ERK1:

IADPEHDHTGFLpTEYVATR ERK1_T202

IADPEHDHTGFLTEpYVATR ERK1_Y204

IADPEHDHTGFLTEYVApTR ERK1_T207

ERK2 Ordered:

VADPDHDHTGFLpTEYVATR ERK2_T185

VADPDHDHTGFLTEpYVATR ERK2_Y187

JNK1/3 Ordered:

TAGTSFMMpTPYVVTR-JNK3_T221;JNK1_T183

TAGTSFMMTPpYVVTR-JNK3_Y223;JNK1_Y185

As correctly pointed out by the reviewer, only the phosphotyrosine peptides were consistently detected. In contrast, the phosphothreonine peptides were only detected sporadically, both in the DDA data used to create the library nor in the SPIED-DIA experiments with the cell line panel (see table below). This is likely due to their poor synthesis yields, as indicated by their more than 10-fold lower intensity. In the label free analysis the peptides were also only identified sporadically.

This observation highlights the point we made about the limitations of our pooled synthesis strategy. We already pointed this out in the Discussion section:

“To economize, we opted for pooled synthesis of heavy reference peptides. However, this approach led to unsuccessful synthesis of some desired peptides, rendering them unusable in our targeted strategy.”

Annotation	Sequence	Library performance (Library BioData), DDA			Identifications in Cell-Line panel experiment (SPIED-DIA)	
		N identified precursors in 3 runs (charge state)	Mean Intensity top 3 precursors (log10)	Mean Andromeda score top 3 precursors	Spike-in confidently identified (#runs/34 total)	Mean Intensity (filter for top 1 precursor per run, log 10)
ERK1_T202	IADPEHDHTGFLpTEYVATR	2	5.06	77.0	0	
ERK1_Y204	IADPEHDHTGFLTEpYVATR	7	6.63	165.	34	5.50
ERK1_T207	IADPEHDHTGFLTEYVApTR	2	5.12	66.4	8	3.89
ERK2_T185	VADPDHDHTGFLpTEYVATR	1	5.32	71.2	0	
ERK2_Y187	VADPDHDHTGFLTEpYVATR	4	6.46	142.	34	5.49
JNK3_T221;JNK1_T183	TAGTSFMMpTPYVVTR	1	5.01	47.6	6	3.88

JNK3_Y223;JNK1_Y18 5	TAGTSFMMTPpYVVTR	14	6.31	153	34	5.40
------------------	----	------	-----	----	------

5. To achieve full activation of ERK and JNK both the threonine and the tyrosine in the activation loop need to be phosphorylated, i.e. -pT-X-pY-. The peptide standards do not include such double phosphorylated peptides, but only single phosphorylated versions. That means what is measured are the single phosphorylated forms that are only partially active. This could be confounding the biological interpretation of the results. For instance, the distribution of single vs. double phosphorylated forms can impact cellular transformation of liver cells [Seehofer et al. Context-specific flow through the MEK/ERK module produces cell- and ligand-specific patterns of ERK single and double phosphorylation. Science Signaling 9 (413), ra13., 2016]. Thus, it is important to measure both single and double phosphorylated MAPK species.

We agree with the reviewer that only the dual phosphorylation of the activation loop fully activates ERK and JNK. However, obtaining a standard peptide with dual phosphorylation is technically challenging (or costly), especially in our pooled synthesis format.

However, as the phosphorylation kinetics of ERK (and potentially also JNK) is not processive, i.e. occurs one phosphorylation at a time (see e.g. [https://www.jbc.org/article/S0021-9258\(23\)02262-7/fulltext](https://www.jbc.org/article/S0021-9258(23)02262-7/fulltext)), the dynamics of singly phosphorylated ERK and JNK is strongly linked to double phosphorylation. Therefore, singly phosphorylated peptides can also provide important information about kinase activation state. Nevertheless, we believe that a key measure for kinase activity is the detection of target phosphosites, as we do by target enrichment analyses.

To acknowledge that phosphorylation of both threonine and tyrosine residues are essential for MAPK activation we added the following sentence to the Discussion section:

“Ideally, such target sets would include peptides from MAPK activation loops that are phosphorylated at both threonine and tyrosine residues, as phosphorylation at both sites is essential for activation.”

6. Fig. 6. All inhibitor concentrations should be given.

We are not sure what exactly the reviewer is referring to since we did provide information about inhibitor concentrations in the Material and Methods section of the manuscript. We now added this information to the legends of Fig. 2 and 3.

7. Supplementary Table 1. It would be useful to list the heavy peptides that were consistently detected separately.

This is now Supplementary Table 2. This table lists the number of heavy identifications per precursor and cell line.

8. While it is common that phosphoproteomics studies only measure phosphopeptides and ignore that changes in phosphorylation could reflect changes in protein abundance, it would increase the biological meaningfulness of the results to normalize the abundance of phosphopeptides against the corresponding non-phosphorylated peptides. Thus, a set of peptide standards that utilize both the phosphorylated and non-phosphorylated form of the peptides would be very useful. The non-phosphorylated peptides could for instance be measured in the unbound fraction of the phosphopeptide enrichment step or in a total protein expression profile.

We fully agree that this would be a useful addition. Adding the non-phosphorylated forms of each target phosphopeptide would even allow calculation of site occupancy, which could be informative. We added the following sentence to the Discussion section:

“Moreover, including non-phosphorylated versions of target phosphopeptides in the spike-in strategy could provide additional insights into changes in protein abundance and phosphorylation site occupancy.”

Minor points

1. The kinase activation loop peptide standards (reference 25) are commercially available: <https://www.jpt.com/spiketides-set-kinase-activation-loops-human-heavy/SPT-KAL-L> Line 530: -20C

Thanks for pointing this out! We included this information in the discussion section:

“A viable alternative is utilizing off-the-shelf reference peptide collections like the PQ500 standard for plasma proteomics (Biognosys), the kinase activation loop peptides collection (JPT) or the recently introduced multipathway phosphopeptide standard (Thermo)⁵⁷”

We kept the sentence about the SigPath and T-loop libraries as is because T-loop library does not appear to be identical to the kinase activation loop peptide collection from JPT.

2. Line 632: 30 minutes instead of 20 (according to results section) **changed in the results section to 20 minutes**

3. For consistency: Caco2 in the whole text and figures (fig. 2 and legend of fig. S9) instead of CaCo2 **done**

4. Line 726: only F-test < 0.1 is used not 0.05 (figures 4 and 5). **corrected**

5. Fig. 3 A & D: suggestion to keep same colors for the different treatments between A and D **corrected**

6. Fig. 3B: y-axis label unclear **changed**

7. Fig. 6 B-D: unit on all x axis - μM instead of uM **changed**

8. Fig. 6 B-D: the ‘drug schematics’ for MEKi and PI3Ki/JNKi are confusing as their coloring and positioning suggests that they are related to what is shown on the y-axis and the color coded lines. Suggest to remove them. **changed**

9. Fig. S3: 'blue the traces' should be 'the blue traces' corrected
10. Fig. S4: 'n-terminal' should be 'N-terminal', and '. modified' should '. Modified' corrected
11. Fig. S5A: 'A' missing in figure corrected
12. Please reorder supplementary figures according to their appearance in text done
13. Figs. 5A and 5C are not mentioned in the text. These were mentioned already, but in general all figure mentions were cross-checked

Reviewer #5 (Remarks to the Author):

Thanks for your help!

Final point by point response

We thank all reviewers for all the work! We have addressed the remaining points as explained in detail below.

Reviewer #1 (Remarks to the Author):

The authors have addressed most of our concerns in a comprehensive and satisfactory way. We appreciate the detailed and well-written discussion of all reviewer comments and the inclusion of our suggested experiments. The newly added negative control nicely demonstrates SPIED-DIAs potential to separate truly from randomly missing peptide detections (Fig. 1I). In the same way, the additional analysis of the titration dataset via Skyline provides interesting insights into chromatographic features of individual targeted peptides (Fig. S5). The differences in quantitation performance between DIA-NN and Skyline are striking and highlight DIA-NN as a software of choice for DIA and, seemingly even more so, for SPIED-DIA experiments (Fig. S3&S4).

A major selling point for SPIED-DIA is its potential to combine targeted quantifications of pre-defined peptides with explorative screens within one measurement. The authors have nicely demonstrated this. One remaining question from our side is the quantitation of phosphopeptides, which were not included into the spike-in panel i.e. which were only analyzed in the explorative part. Is their quantification in DIA-NN in any way influenced by the presence of the heavy spike-in panel?

We tested whether phosphopeptides not in the heavy spike-in panel (those measured only in the exploratory portion) are quantitatively affected by the presence of the heavy panel in DIA-NN. To do this, we compared the coefficient of variation (CV) across technical triplicates for phosphopeptides measured with vs. without the 50 fm heavy spike-in (the samples from the benchmarking experiment reported on in Figure 1G).

The results are:

Mean CV with spike-in: 0.0162

Mean CV without spike-in: 0.0203

Difference: 0.0041 (95% CI: -0.00446 to -0.00379; $p < 2.2 \times 10^{-16}$)

So, a small but statistically significant reduction in CV can be observed when the heavy spike-in is present. In practical terms, this suggests that phosphopeptide quantification is indeed slightly influenced by the presence of the spike-in peptides, but the effect size is relatively small.

Minor comments are:

- In Figure 1H, please include a green icon for E.coli to visualize what the green portion in the tubes represents.

Done

- Please show intensities for 0ng in the left part of figure 1I

This is not possible because there are no light identifications in the 0ng condition, see plot below:

After these smaller concerns have been addressed, we fully recommend this paper for publication and congratulate the authors to this nice work.

Reviewer #2 (Remarks to the Author):

Thank you very much!

Reviewer #3 (Remarks to the Author):

With respect to your response about the novelty of the method, I agree that it is unclear in Litichevskiy et al. exactly how all data were acquired. However, in another paper that utilized the same data, Vaca Jacome et al. (Nat. Method 17:1237-44, 2020) show how P100 DIA mode data could be interrogated for endogenous analyte counterparts to spiked-in phosphopeptide standards. This demonstrates that the method had been conceived and reduced to practice previously. In that work, the authors were able to achieve “quantification of 92% of all peptides with data completeness of 79%” using their workflow (Fig. 3). Shouldn't that be the benchmark to meet for novelty and/or improvement?

Again, my concerns with this manuscript were not due to the quality of the experiments. They were more about the novelty vis a vis publication in Nature Communications, and whether this particular workflow's efforts justified its return. I applaud the extensive revisions made by the authors to address my other points, and I feel that those revisions have made their conclusions stronger. But not more novel or impactful. Ultimately it is up to the editorial staff to make a judgement about novelty and impact.

We thank the reviewer for the thoughtful and supportive feedback.

With regard to the novelty of the method we feel that we did cite the relevant previous work extensively and clarified that our method builds on previous achievements and is thus not entirely new.

Reviewer #3 (Remarks on code availability):

The code seems OK, but it seems to me that several input files required to replicate the figures in the paper are missing from the code base. So it's not particularly useful from a reproducibility standpoint. I can not comment on the “keyscripts” portion as I do not have test data available to verify functionality of the code.

We have made the scripts available. Please find the details for specific figures below.

Figure 1DE, Supplementary Figure 1ABC: Raw data and DIA-NN output: on Pride (including spectral library) Code to analyse/visualise from DIA-NN output: on Github under GitHub_SPIED-DIA/Manuscript_scripts/Technical_Benchmark1/FullPhosphoDil.R Source Data Figures uploaded.

Figure 1F: SupplementaryData1 for an overview of the selected phosphosites

Figure 1G, Supplementary Figure 1D: Raw data and DIA-NN/MaxQuant output are available on Pride (including spectral library)

Figure 1HI, Supplementary Figures 3,4,5: Raw data (Rev_Benchmark3_Rawfiles.zip) as well as necessary input files (Rev_Benchmark3_DIANNoutput.zip, DIA-NN report) and JPT library data (on Pride under Lib_Benchmark3.csv, id in GitHubscript: 20241104JPTlib_noNA.csv) are in the associated PRIDE repository. Code to analyse/visualise from DIA-NN output: on Github under GitHub_SPIED-

DIA/Manuscript_scripts/Technical_Benchmark3/Analysis_BM3.R Source Data Figures uploaded.

Figure 2, supplementary figure 6: Raw Bio-plex data (Supplementary Data 3), as well as source dat for the figures is provided

Figure 3CD: Raw data and DIA-NN output (including spectral library files) are available on PRIDE (PanelGFmix_Rawfiles.zip, PanelGFmix_outputDIANN_LF.zip). Script to analyse the DIA-NN output and generate PCA plots is available under GitHub_SPIED-DIA/Manuscript_scripts/Biological/CellLinePanel_Global/CellLinePanel_Global_Norm.R

Figure 3/4 (SPIED-DIA): Raw data and DIA-NN output (including spectral library files) available on PRIDE (PanelGFmix_Rawfiles.zip, PanelFGmix_outputDIANN_SPIED.zip). Intermediate Analysis files have been attached as Supplementary Data 4-7. Source Data for Figures 4ABCE, Supplementary Figure 9 is uploaded. Script to analyse the DIA-NN output and visualize is available under GitHub_SPIED-DIA/Manuscript_scripts/Biological/CellLinePanel_Targeted/CellLinePanel_TargetedAnalysis.R

Figure 5: Raw data and DIA-NN output (including spectral library files) are available on PRIDE (PanelGFmix_Rawfiles.zip, PanelGFmix_outputDIANN_LF.zip). Intermediate analysis files have been attaches as Supplementary Data (Supplementary Data 9-14). Script to analyse the DIA-NN output and visualize is available under GitHub_SPIED-DIA/Manuscript_scripts/Biological/CellLinePanel_Global/CellLinePanel_Global_Norm.R Script to perform differential expression analysis with limma as well as generate supplementary Figures 10, 11 and 12 is available under GitHub_SPIED-DIA/Manuscript_scripts/Biological/CellLinePanel_Global/DiffExpression.R (requires HpHlibrary annotation file, provided as supplementary Data 12) Script to generate Figure 5A is available under GitHub_SPIED-DIA/Manuscript_scripts/Biological/CellLinePanel_Global/Cluster_Kinase_Enrichment.R Script to generate Figure 5C is available under GitHub_SPIED-DIA/Manuscript_scripts/Biological/CellLinePanel_Global/Cantleykinase_analysis.R Source Data for Figure 5C uploaded (as well as intermediate analysis files, Supplementary Data 13, 14).

Figure 6: Raw Data available as Supplementary Data 15 and 16.

Supplementary Figures 15, 16: Raw Data and DIA-NN output available on PRIDE

Reviewer #4 (Remarks to the Author):

The authors have addressed all my comments and also the comments of the other reviewers satisfactorily in their revised paper. I am happy to recommend the paper for publication.

Thanks a lot for the supportive feedback!

Reviewer #5 (Remarks to the Author):

Thanks a lot for your help!